



# Subsurface structures of a quick-clay sliding prone area revealed using land-river reflection seismic data and hydrogeological modelling

Silvia Salas-Romero[1], Alireza Malehmir[1], Ian Snowball[1], and Benoît Dessirier[1,2]

[1]Department of Earth Sciences, Uppsala University, Uppsala, 75236, Sweden
[2]Department of Physical Geography, Stockholm University, Stockholm, 10691, Sweden

*Correspondence to*: Silvia Salas-Romero (silvia.salas_romero@geo.uu.se)

**Abstract.** Quick-clay landslides are common geohazards in Nordic countries and Canada. The presence of potential quick clays is confirmed using geotechnical investigations, but near-surface geophysical methods, such as seismic and resistivity
surveys, can also help identifying coarse-grained materials associated to the development of quick clays. We present the results of reflection seismic investigations on land and in part of the Göta River in Sweden, along which many quick-clay landslide scars exist. This is the first time that such a large-scale reflection seismic investigation has been carried out to study the subsurface structures associated with quick-clay landslides. The results also show a reasonable correlation with the radio magnetotelluric and traveltime tomography models. The morphology of the river bottom and riverbanks, as e.g. subaquatic
landslide deposits, is shown by side-scan sonar and bathymetric data. Undulating bedrock, covered by subhorizontal sedimentary glacial and postglacial deposits is clearly revealed. An extensive coarse-grained layer exists in the sedimentary sequence and is interpreted and modelled in a regional context. Individual fractures and fracture zones are identified within bedrock and sediments. Hydrological modelling of the coarse-grained layer confirms its potential for transporting fresh water infiltrated in fractures and nearby outcrops. The groundwater flow in the coarse-grained layer promotes leaching of marine
salts from the overlying clays by slow infiltration and/or diffusion, which helps in the formation of potential quick clays. Magnetic data show coarse-grained materials at the landslide scar located in the study area, which may have acted as a sliding surface together with quick clays.

## 1 Introduction

Quick clays are sensitive glacial and postglacial sediments, mostly deposited in a shallow-water marine environment, whose
structure can collapse and liquefy if disturbed (Osterman, 1963; Torrance, 2012). These sediments were deposited in the last deglaciation and early postglacial, and subsequently isostatically raised above sea level (Torrance, 2012). Due to the infiltration of meteoric waters, mineral salts were leached out, which changed the salinity of the pore water and altered their soil properties (Rosenqvist, 1953). Leaching of salts is important in the development of the characteristic quickness behaviour (Rosenqvist, 1946; Torrance, 2012), but there are other factors that influence the formation of quick clays, such as





the presence of dispersing agents and pH level (Salas-Romero et al., 2015; Torrance, 2012). The presence of quick clays can only be confirmed using geotechnical site and laboratory investigations (Rankka et al., 2004), estimating the sensitivity, i.e. is equal to the ratio of undrained undisturbed to remoulded shear strength. In Sweden, quick clays are defined as clays with sensitivity higher than 50 and remoulded shear strength of less than 0.4 kPa. Quick-clay landslides are common in northern

countries such as Sweden, Norway and Canada. They are world known from catastrophes such as in Rissa (Gregersen, 1981), Tuve (Larsson and Jansson, 1982), and Saint-Jude (Locat et al., 2017), and they have been studied in different fields like geotechnics, geophysics or geology (Dahlin et al., 2013; Lundström et al., 2009; Salas-Romero et al., 2015; Sauvin et al., 2014; Solberg et al., 2016; Wang et al., 2016).

In order to define areas susceptible to quick-clay landslides in Sweden, Rankka et al. (2004) reviewed a number of

geological and geohydrological prerequisites for the formation of quick clay in nature. This list included: glaciomarine sediments, thin clay deposits, underlying coarse-grained layers, peaks in the bedrock surface that retain accumulating groundwater, artesian groundwater pressure, highly permeable layers within the clay deposits, height above sea level, organic soils, a large catchment area, and infiltration from more than one direction. A combination of several near-surface geophysical methods, such as reflection seismic, radio magnetotellurics (RMT), and traveltime tomography can be used to

identify some of these prerequisites, e.g. the presence of the underlying coarse-grained layer or undulations in the bedrock surface. Estimation of the catchment area and type of leaching process (fresh water percolating through the deposits, artesian, and diffusion) requires hydrological modelling based on borehole data and the elevation surfaces of bedrock and coarse-grained layer (top and bottom), obtained from the integration of geophysical, geotechnical and topographical data.

Although quick-clay landslides are influenced by the geological structures, groundwater conditions and soil type, surface

topography and geomorphology also play an important role in making an area susceptible to landslides (Wang et al., 2016). The Göta River valley, which is located in southwest Sweden, hosts many landslide scars, most of which are quick-clay related (Swedish Geotechnical Institute–SGI, 2012a). The valley is filled with thick marine postglacial deposits that overlie undulating bedrock (SGI, 2012a). The Göta River is one of the longest rivers in Sweden (93 km long), the source of drinking water for more than 700,000 people, a transport route, and is the main outflow of the country's largest lake, Vänern (SGI,

2012a). One or more large landslides could dam the river, affecting river-based transportation and have economic and social consequences. In June of 1957 a landslide took place at the sulfite factory located next to the Göta River (in the southern part of Lilla Edet). This landslide, one of the biggest to have been observed in Sweden, caused a lot of structural damage (Fig. 1) and the loss of three lives. The landslide propagated backwards in a retrogressive manner and along the riverbank, transporting large amounts of materials into the river, triggering a wave of 5 to 8 m height and covering around 32 hectares

(Hultén, 2006; Odenstad, 1958). It has been suggested that the landslide was triggered by the infiltration of sulfite liquor and other chemicals into the ground, which reduced the clay shear strength, in combination with other factors such as the erosion at the riverbank and the presence of quick clay (Odenstad, 1958).

Triggering of quick-clay landslides is influenced by natural conditions (heavy rainfall, high-low water flow along the river, river erosion or variation of the groundwater level) and by human activities (constructions or loading/unloading works)



(Thakur et al., 2014). With a minimal slope angle and a place to flow to, any mechanism that increases stress on the quick clays or reduces their strength may trigger a landslide. The model scenarios for climate change over the next hundred years present a warmer and wetter Sweden (Swedish Government Official Reports, 2007), which means more precipitation in the west of the country, more runoff water and higher river discharge, increasing the likelihood of landslides in areas that are

already prone to them. More variability in river level, as predicted by the climate studies (Swedish Government Official Reports, 2007), may destabilize some slopes along the shores, causing new landslides.

The work presented in this paper is a continuation of previous studies done in an area prone to quick-clay landslides in southwest Sweden in 2011 and 2013 (Malehmir et al., 2013a, Salas-Romero et al., 2015). The study area is within the municipality of Lilla Edet in a region called Fråstad, crossed by the Göta River. Lilla Edet has around 14,000 inhabitants

(Lilla Edets Kommun, 2017), and is located approximately 8 km south of this area, on the eastern side of the river. The geology at the study area consists mostly of gentle reliefs of glacial and postglacial deposits such as clay, silt and sand, with some granite to granodiorite bedrock outcrops (© Geological Survey of Sweden–SGU, Fig. 2a). Along the shorelines, landslide scars can be found. Precisely at the survey site several landslide scars are visible along both sides of the river, one of them being of particular interest due to its location in the middle of the survey area (Fig. 2a–b). A net of morphological

lineaments, mostly fracture zones, covers most of the area represented in Fig. 2a. Among these lineaments, one of the prominent ones follows the river profile and another one has an E-W direction. It is known that the bedrock in the Göta River valley has an extended system of cracks, with a fault zone that follows the river channel (SGI, 2012b). Figure 2b shows high-resolution LiDAR data (© Lantmäteriet) ranging from 7 to 97 m elevation. A couple of seismic profiles acquired in this survey, lines 6 and 7 are located in an area with lower elevation compared to the rest of the land seismic lines. At this

position a large landslide scar is visible (Fig. 2a–b) that shows a horst and graben pattern, classifying this landslide as spread type (Demers et al., 2107).

Previous studies included P- and S-wave reflection and refraction surveys, potential fields, controlled-source tensor (CSTMT) and RMT, electric resistivity tomography (ERT), and ground-penetrating radar (GPR). In 2013 three boreholes (BH1 to BH3 in Fig. 2a–b) were made in the study area primarily to ground truth geophysical interpretations but also to

collect undisturbed core samples for laboratory measurements. Geophysical, geotechnical and borehole data show that a coarse-grained layer underlies leached clays (potential quick clays) and quick clays in some places within the study area. This layer plays a role in leaching the marine salts from the overlying clays and speeds up the formation of quick clays. Some geotechnical investigations (Löfroth et al., 2011) show that when the coarse-grained layer is thicker, the thickness of the quick clays is also larger. The sediments above the coarse-grained layer are intercalating layers of silt and clay, and

below they are mostly marine clays that extend down to bedrock. The studies also suggest the eastern part of the study area has higher proportion of leached clays than the western part.

This research is based on a joint interpretation of multidisciplinary datasets for (i) 3D geological/geophysical modelling of the larger-scale subsurface structures overlying and existing within bedrock – like possible fracture zones, (ii) understanding the role of the identified coarse-grained layer and its spatial relationship with the bedrock surface that may improve the





hazard assessment, (iii) hydrological modelling of the groundwater within the coarse-grained layer to better understand the development of quick clays in the study area, and (iv) investigating the riverbanks of the Göta River, its bed and mass-movement deposits. Reflection seismic, P-wave refraction tomography (Wang et al., 2016), and resistivity models (Bastani et al., 2017; Wang et al., 2016) are correlated with borehole data (Branschens Geotekniska Arkiv–BGA, 2018; Salas-Romero

et al., 2015) for the identification of different types of clays, coarse-grained materials and bedrock. Using the interpreted seismic sections together with total sounding (BGA, 2018) and high-resolution LiDAR data (© Lantmäteriet), elevation surfaces from the top of bedrock and top and bottom of the coarse-grained layer are modelled. This study shows not only that this layer probably covers a larger area than initially thought (earlier studies showed the local extension of this layer), but also confirms its hydrological potential as a transport path for infiltrating fresh water from nearby outcrops and fractures.

Magnetic surface data serve for illustrating that the coarse-grained layer together with quick clays may have acted as sliding surface at the landslide scar located within the survey area. Other surface data, such as side-scan sonar and bathymetry, are also analysed to investigate the riverbed and their influence on the development of quick-clay landslides. This work provides a good example of the integration of a large amount of different types of data for the study of an area prone to quick-clay landslides.

## 15  2 Data acquisition

### 2.1 Land reflection seismic

Four land reflection seismic profiles were acquired during two weeks in 2013 (lines 2b, 5b, 6 and 7 in Fig. 2a–b) totalling a length of approximately 3.8 km, aiming to complement existing reflection seismic data (Lundberg et al., 2014; Malehmir et al., 2013a, 2013b) and to obtain an overall understanding of the larger-scale structures. The new lines extend the study area

along the N-S direction as line 5b crossed to the northern side of the river. Table 1 compiles the main acquisition parameters. In three of the lines, 2b, 6 and 7, the geophone and shot spacing were the same (4 m), while in the longest line, 5b, they were different. In the southern part of line 5b cabled geophones of 28 Hz were used every 4 m, and in the northern part wireless stations were deployed every 10 m, alternating single-component (1C) receivers of 10 Hz and three-component (3C) broadband digital MEMs (micro-electro mechanical systems) sensors. Due to the great length of line 5b (~2300 m),

dynamite was used as the source energy and fired at every 20 m in order to obtain high-quality data. In line 2b an accelerated weight drop was used, and in lines 6 and 7 a sledgehammer was the main seismic source because the accelerated weight drop experienced technical problems. At the beginning and at the end of lines 6 and 7, extra shots were made using dynamite. These two lines were connected to each other during the survey, which allowed simultaneous data acquisition along both lines. Due to the large distance between them (~300 m) the initial idea of a joint 3D first-break tomography to resolve the

bedrock surface was not possible, nevertheless individual results were obtained for each line (Wang et al., 2016). The acquisition system was Sercel 428$^{TM}$, and the survey coordinates were obtained using a differential global positioning system (DGPS) with high-precision geodetic data. The range of lateral and vertical resolutions for this method is shown in Table 2,





together with the spatial sampling information from other methods used in this work. The shown lateral resolution for the reflection seismic data has been obtained calculating the first Fresnel zone at 20 m depth.

## 2.2 River reflection seismic

Reflection seismic data along the Göta River were acquired from the vessel 'Ocean Surveyor' (property of SGU) in 2000, and were made available as raw shot records. Two types of acquisition were registered: single-channel (3.5 kHz echo sounder) and six-channel (Fig. 2a–c). Both lines have similar length (around 16.9 km) and run parallel, with their initial and final points very close to each other (they are shifted with respect to each other by approximately 11 m). In the single-channel line the receiver and shot positions were the same (the average distance between consecutive points is 3 m). In the six-channel line the distance between the source and the nearest receiver is 6 m (maximum offset is 21 m), the receiver spacing being 3 m. A 10 in$^3$ sleeve gun was used as source. The frequency range is between 100 and 1000 Hz. Table 2 shows the lateral and vertical resolution for this method, which have been calculated following the same procedure as in the land reflection seismic data.

## 2.3 Magnetics

The ground magnetic data were collected by Uppsala University during the field campaign of 2011 (Malehmir et al., 2013a). The purpose of this survey was to delineate the bedrock topography by estimating the changes in the magnetic field generated by the rock magnetism. The total-field magnetic and vertical gradient were measured using a walking mode GPS-mounted magnetometer during five days. During the first three days a N-S direction was followed and during the last two days an almost east-west direction. In total, 17128 points were surveyed (see spatial sampling in Table 2). A base station was used for correcting the diurnal variations and instrumental drift. The position of this base station changed along the five days, with the biggest difference between the first and the rest of the days (around 53 m). The vertical gradient data did not provide convincing results and thus were disregarded for detailed studies.

## 2.4 Side-scan sonar and bathymetry

The side-scan sonar data on the Göta River were acquired by SGU in 2000, and were available as a 2D georeferenced image file (see Table 2). The system used was a klein of 500 kHz. These data are obtained by transmitting sound waves, which are then received as reflected sound waves from underwater elements, and allow to produce an acoustic image of the materials and morphology of the riverbed (Kaeser et al., 2013). The amplitude values of the image are represented with a grey scale that indicates the strength of the reflectivity and possible density of the materials. Normally, dark areas are considered coarse materials like gravel, and light areas are considered fine materials like clays. The side-scan sonar data cover most of the river shown in Fig. 2a–b, and extending a bit further to the south.





Marin Miljöanalys AB collected the bathymetric data using multi-beam echo sounding (Kongsberg EM3002-D, 300 kHz) in 2009 under the assignment of SGI (Marin Miljöanalys AB, 2009). The goal was to create a high-resolution topography model of the riverbed. The data were available as a georeferenced file (see resolution in Table 2).

## 2.5 LiDAR

The LiDAR scan was collected by Lantmäteriet in 2011. The survey was done from a height of 2000 m and the average point density is from 0.5 to 1 points/m$^2$ (see resolution in Table 2).

## 3 Reflection seismic processing

Table 3 presents the main processing steps for the land and river reflection seismic data. The processing of the land reflection seismic data was similar for lines 2–2b, 6 and 7. The preparation of the data required zero-time correction, vertical

stacking of repeated shot records, as well as merging of the new line 2b with the 2011 line 2 (Malehmir et al., 2013b). Removal of first arrivals using a carefully designed top mute filter using picked first breaks and the application of stretch mute (Schmelzbach et al., 2005) helped to enhance the reflections at the shallow parts and avoid misinterpretation of the first arrivals. Refraction static corrections did not give satisfactory results for any of the lines, and they were not applied further. Elevation static corrections were, however, applied using the highest elevation as datum and a velocity of 1500 m/s. As the

data still looked noisy and with lower resolution, more preprocessing steps were necessary. Deconvolution before stacking helped in obtaining a reasonably clear seismic section. A series of constant velocity stacks (from 800 to 4000 m/s) was used in order to obtain the most coherent bedrock reflections. A post-stack $fk$-filter and surface-consistent residual static corrections were applied for data along lines 6 and 7 for improving the continuity of the reflections. Black et al. (1994) show that the migration process is not really necessary for near-surface seismic imaging applications although it can reduce the

noise level. After a series of tests we eventually concluded that migration did not lead to any improvement because the reflections are mostly subhorizontal or gently dipping.

Figure 3 shows an example of a shot record from line 7 (SH7, see the position in Fig. 2a). Figure 3a presents the raw shot gather with only trace balance applied, and Fig. 3b presents the preprocessed shot gather after elevation static corrections, Wiener deconvolution, band-pass filter, trace editing, removal of first breaks and trace balance. The bedrock reflection, B1,

is visible already in the raw data but improved in the preprocessed shot record.

The processing of the land reflection seismic data for line 5–5b was slightly different from the rest of the lines. First, the wireless data needed to be resampled from 1 to 0.5 ms to be consistent with the cabled geophone data. Then, the cabled geophone, 1C and 3C wireless (vertical component) data were merged. Once all the data were joined, a delay of around 1 s between the cabled geophone and the wireless part was observed (the wireless data were shifted up 1 s). With the data zero-

time shifted, the next step was to merge them with those from the 2011 line 5 (Malehmir et al., 2013b). As the receiver distance was different for the cabled geophone and the wireless parts, it was necessary to process each part separately,



applying different geometries for each case (common depth point, CDP, spacing equal to 2 m in the south and 10 m in the north). Before velocity analysis, both processing included elevation static corrections, removal of first arrivals using a top mute filter (surgical mute for the wireless data), band-pass filter, spectral whitening, and *fk*-filter for the wireless data. The high-quality data for line 5–5b allowed a relatively simple processing flow, where the most important step was the velocity

analysis (performed at 10-20 m lateral spacing in the southern part; in the northern part constant velocities, from 800 to 4000 m/s, were tested). The velocity analysis of the wireless seismic data revealed that the deeper part of the section needed higher velocities to obtain visually coherent reflections, thus the data were divided in two parts for processing: from 0 to 80 ms and from 80 to 500 ms. The post-stack processing included band-pass filter for both types of data (cabled geophone and wireless), and post-stack deconvolution in the case of the wireless data.

Figure 4 shows an example of a shot record along line 5–5b (SH5, see the position in Fig. 2a). Figure 4a is the raw shot gather, and Fig. 4b is the preprocessed shot gather (elevation static corrections, band-pass filter, spectral whitening, trace editing, removal of first breaks and automatic gain control, AGC). In Fig. 4b a number of reflections seem to be revealed; bedrock reflection B1 and a shallower one from a coarse-grained layer S1. The sediments show P-wave velocities ranging from 1000 to 2000 m/s, while bedrock shows velocities much higher than 3000 m/s (Wang et al., 2016). The values are

similar to those in line 7 (Fig. 3a), except for the direct wave, which is much slower in that case. It may be related to near surface effects or differences in topography.

The processing of the river reflection seismic data was simpler compared to the land seismic processing (Table 3). In the case of the single-channel data only Wiener deconvolution was applied for removing multiples as much as possible. The six-channel data required the creation of marine geometry according to the receiver and shot positions. A CDP spacing of 1.5 m

was used for making the geometry. For the six-channel data more processing steps were necessary, where Wiener and post-stack deconvolution helped to improve the final results.

Land and river reflection seismic data were time to depth converted using a constant velocity of 1500 m/s. This value was justified based on the available borehole data for depth calibration, although a slight error on the order of 1-3 m can still be expected.

**4 Results and interpretations**

**4.1 Land seismic lines**

Figure 5a–d shows, from top to bottom, the seismic results for line 2–2b, the interpreted horizons and structures, i.e. S1, B1 and F, and the P-wave refraction tomography and RMT resistivity results obtained in earlier studies (Wang et al., 2016). Figure 5b also includes the natural gamma radiation and magnetic susceptibility data from borehole BH1 (Salas-Romero et

al., 2015), and the total sounding data from borehole 7065 (BGA, 2018). S1 is a subhorizontal layer only identified in the southeastern part of the line; the strong decrease in the gamma log of BH1 (distance to the seismic line 13.4 m) and the increase of the magnetic susceptibility coincide with this interface. S1 is a coarse-grained layer previously identified in



Salas-Romero et al. (2015). In the northwestern part of the line, next to the river, the S1 reflection is not visible. B1, interpreted as the top of bedrock (BH1 reached the top of bedrock, see Salas-Romero et al., 2015), is more irregular and has a higher amplitude reflection than S1. B1 shows a clear undulating morphology, reaching the ground surface at approximately CDP 525, and dipping towards the river in the northwestern side; in borehole 7065 (distance to the seismic

line 0.02 m) the strong increase in the total sounding curve happens at bedrock depth, not showing any important change at the suspected position of S1. The discontinuous reflectivity at bedrock level indicates the presence of fractured or disturbed materials (F) in the northwestern side. The P-wave refraction tomography results (Fig. 5c) indicate, in general, high velocities (>4000 m/s) below B1 and lower velocities in the overlying sediments (mostly between 350 and 3000 m/s). The top of bedrock is well delineated by high velocities with some exceptions at the extremes of the line, which may be due to

the lack of ray coverage. The S1 horizon shows for the most part higher velocities (1500-3000 m/s) compared to the sediments above and below this layer in the southeastern part of the line. The resistivity results only cover part of the line (Fig. 5d); above the thin dashed white line, the RMT model is well resolved with high confidence. Wang et al. (2016) estimated the penetration depth (thin dashed white line position) using a method shown by Spies (1989), and the same criterion is followed for the rest of the lines. Low resistivity values (between 3 and 100 Ωm) are observed above B1, and

higher values (>100 Ωm) below, with very high at the position closer to the surface (up to 1000 Ωm, some outcrops are close to this location, see Fig. 2b). At the S1 position the values are around 80-100 Ωm, which agrees with the material classification (coarse sediment) given in Solberg et al. (2012). The values immediately above S1, between 10 and 80 Ωm, may indicate silt or leached clay deposits–potential quick clays (Solberg et al., 2012). Quick clay was identified above the coarse-grained layer during the visual inspections of the core samples of boreholes BH1 to BH3 (Salas-Romero et al., 2015).

Figure 6a–e shows the seismic results for line 5–5b, their interpretation, the earlier P-wave refraction tomography and RMT resistivity results (Wang et al., 2016), and the airborne transient electromagnetic (ATEM) resistivity results (Bastani et al., 2017). Figure 6b also includes the natural gamma radiation and magnetic susceptibility data from borehole BH3 (Salas-Romero et al., 2015), and the total sounding data from borehole 7062 (BGA, 2018). Note that the quality of the seismic data is different on each side of the river, due to the lower sampling in the northern side (10 m). Nevertheless, the delineation of

S1 and B1 is possible along the whole line. The S1 horizon shows continuity along the line, except between CDPs 400 and 480. At these positions S1 is not visible, likely due to lower fold in the seismic data (see fold distribution along line 5–5b in Fig. 6a). Other possibilities cannot be disregarded, e.g. bedrock movement and/or fractures in the bedrock, and/or deposits disturbed by human activities such as excavation works (a trench runs perpendicular to the line at this position, although neither its width nor its depth seem to be of the same size of the observed anomaly in the seismic data). A fracture (F) can be

inferred in the bedrock at around CDP 440 with some diffraction signatures suggesting the presence of a strong bedrock curvature or edge. The biggest changes in the gamma and magnetic susceptibility logs in BH3 (distance to the seismic line 0.23 m) and in the total sounding curve in 7062 (distance to the seismic line 90.3 m) coincide with the depth of S1. The undulating B1 reaches close to the surface around CDP 500 and dips to the river after this point; no clear bedrock dipping is observed in the opposite northern shore data. Below the landslide scar displacement and oblique translation of some





reflections are observed; the sediments appear to have slid towards the river. The top of bedrock at the river position may be at a depth of around 100 m. The P-wave refraction tomography results (Fig. 6c) in the southern side indicate, in general, high velocities (>3000 m/s) below B1 and lower velocities in the overlying sediments (between 300 and 1500 m/s). In the northern side of the line, the tomography results and reflection B1 agree in the northernmost part of the profile, with high

velocities (>3000 m/s) below B1. From the river to CDP 1000 individual high-velocity anomalies (>4000 m/s) are visible above B1, which may be related to the presence of boulders. The S1 horizon shows higher velocity than the sediments around it in the northern part of the line, which is similar to what is observed in Fig. 5c for line 2–2b. In contrast, the southern part of the line does not show any velocity difference at the S1 interface. The RMT resistivity values (Fig. 6d) are mostly high (>100 Ωm) below B1 although the northern side does not have data below the top of bedrock. Above B1, the

RMT resistivity values are low (between 1 and 10 Ωm) except along the S1 horizon, where the values reach >100 Ωm. The values immediately above S1 may indicate leached clay deposits–potential quick clays (Solberg et al., 2012). The ATEM resistivity values (Fig. 6e) do not to delineate the interface B1 as well as the RMT resistivity results in the southern part of the profile, except at the position where the bedrock is closer to the surface. In the northern part of line 5–5b the penetration depth of the ATEM model does not reach the position of B1. In comparison, the ATEM resistivity results seem to delineate

the interface S1 between CDPs 500 and 700, and 880 and 1000. Where there is coincidence, the range of ATEM resistivity values is similar to the range of RMT resistivity values mentioned earlier. Malehmir et al. (2016) interpret a possible fault (low resistivity values sandwiched between high resistivity values identified as bedrock) on the southern shore of the river, between CDPs 730 and 750 approximately at 30 m below sea level.

Figure 7a–c shows the seismic results for line 6, their interpretation and the earlier obtained P-wave refraction tomography

results (Wang et al., 2016). In this case only B1 is identified in the seismic section. The bedrock shows undulating morphology, dipping to the west towards the river, and reaches the ground surface at the end of the line in the eastern side. Between CDPs 100 and 200 from -50 to -100 m elevation (Fig. 7b), a different reflection pattern (dipping to the east of the line) is present, which may be related to 3D effects caused by the rough bedrock topography. Fractured or disturbed materials (F) are identified in the western side of the line and also at around CDP 260. The F materials located closer to the

river seem to coincide with a probable fracture zone present in Fig. 2a, which runs parallel to the river profile. The velocities (Fig. 7c) are, in general, high (>3000 m/s) below the B1 reflector, except between CDPs 150 and 200 where the velocity is high (>3000 m/s) above B1.

Figure 8a–d shows the seismic results obtained along line 7, their interpretation, and the earlier obtained P-wave refraction tomography and RMT resistivity results (Wang et al., 2016). Figure 8b also includes the total sounding data from boreholes

7073 and 7075 (BGA, 2018). B1 is delineated along the line, having the similar appearance as in line 6 (Fig. 7b). The S1 reflection is only delineated between CDPs 100 and 210. Boreholes 7073 and 7075 (distance to the seismic line 29.3 and 1.6 m, respectively) located in the western side of the line show strong increases in their curves at the S1 interface. Some fractured or disturbed materials (F) are identified at around CDPs 160 and 200. The P-wave refraction tomography results (Fig. 8c) are similar to the ones shown in line 6 (Fig. 7c). The velocities are high in the eastern side (>3000 m/s) below B1





and low in the overlying sediments (approximately between 450 and 2000 m/s). In the western side of the line the ray coverage is poor at the depth of B1. The tomography results do not show any velocity difference at the S1 interface. The resistivity values are high (>100 Ωm) below B1 in the eastern side (Fig. 8d); the western side does not have resistivity results below B1 and only partial results above it. The resistivity values are around 10 to 100 Ωm at shallower depths, except at the

S1 position and below it where the values are lower, between 1 and 10 Ωm. In terms of resistivity, these values do not indicate coarse sediments but unleached marine clay deposits (Solberg et al., 2012). Nevertheless, the resistivity model is not well resolved below 10 to 30 m depth along the line.

## 4.2 River seismic lines

Figure 9a and 9b shows the seismic processing results for the single-channel data (© SGU), and their interpretation,

respectively. Figure 9b also includes the total sounding data from boreholes 11014, 11034 and 11094 (BGA, 2018). The seismic results still show many multiples along the line, e.g. between 2000 and 3500 m distance or between 9000 and 11000 m distance. Seven filled channels labelled by C (Fig. 9b) can be distinguished along the line, the larger ones up to 2000 m wide are found in the first 11000 m distance, and the smaller ones are around 1000 m wide. The curves for each borehole show strong increases when intersecting the filled channels, thus indicating the presence of coarse materials within the

sediments. Note that most of the depressions along the bathymetric profile coincide with the channel positions. These depressions can reach up to 10 m difference in height with respect to peaks or mounds that are found between them. Only a peak of the bedrock interface (B1) is interpreted between 5500 and 6000 m distance, separating two adjacent channels. The interpretation of the areas separating adjacent channels in the rest of the cases, at distances around 2800, 8000, 11500, 13000 and 15000 m, is complicated because no more structures are clearly visible. The reasons may include the bedrock being very

close to the surface and/or the presence of fracture zones as shown in the interpretation of the six-channel data.

The results of the six-channel data collection (© SGU) and their interpretation are presented in Fig. 10a and 10b, respectively. Figure 10b also includes the total sounding data from boreholes 11014, 11034 and 11094 (BGA, 2018). The difference in height between the valleys and peaks reaches up to 15 m (Fig. 10a). Although the resolution of the six-channel data compared to the single-channel data is lower, geological features can be distinguished at greater depth (>100 m). In the

interpreted section (Fig. 10b) the same channels (C) identified in the single-channel data (Fig. 9b) can also be delineated, as well as the bedrock highs between 5500 and 6000 m distance (there is a difference of about 10 m in height between both data sets at this point, probably due to the distance between the lines). Strong variations in the borehole data coincide with the filled channel positions. The bedrock undulates and presents several fracture zones between CDPs 800 and 1000, 1350 and 1400, 2350 and 2500, and 2950 and 3000. These fracture zones coincide with fracture zones identified using the geological

information provided by SGU (Fig. 2a). Along the whole line fractured or disturbed materials (F) can be identified at the shallowest sediments, and also at bedrock level. In Fig. 10b we can also observe that the reflection amplitude decreases between CDPs 3000 and 5402, the deeper areas being more affected. The bedrock interface (B1) may be closer to the surface at these positions, thus the low-amplitude region would represent the transparent crystalline bedrock.



Figure 11 shows a detailed section of the river seismic data (© SGU) between CDPs 1800 and 3300. The portion of side-scan sonar data (© SGU) corresponding to the profile AA' (Fig. 11b–c) is presented in Fig. 11a. Figure 11b and 11c are the interpreted sections of single- and six-channel data. Line 5–5b crosses a fracture zone. In Fig. 11a hummocks and disturbed riverbed are observed at the centre of the river bottom; these may be interpreted as landslide debris. The shade colour and the texture indicate denser and coarser deposits, respectively, compared to their surrounding materials. At the position of line 5–5b, a landslide scar can be found in the southern side of the river (see also Fig. 2a and 6a). In the opposite riverbank two more landslide scars are found as well as one gully flowing from the south and a tributary flowing from the southeast (Fig. 2a–b). Based on Fig. 11a, it is unclear whether the deposits at the river bottom originate from the landslides or are fluvial sediments.

# 5 Discussion

## 5.1 Comparison with previous studies

Previous findings established that the formation of quick clays in the study area is influenced by the presence of underlying coarse-grained materials (Löfroth et al. 2011; Malehmir et al., 2013a, 2013b; Salas-Romero et al., 2015; Wang et al., 2016). The coarse-grained layer serves as a path for the leached substances to be transported towards the river. Salas-Romero et al. (2015) mention that the presence of this layer expedites the development of quick clays: infiltration of surface water through outcrops and fracture zones allows relatively fresh water to reach the glacial and postglacial sediments, which leads to leaching of salts and promoting quickness. The borehole information available in the area (BGA, 2018; Löfroth et al. 2011; Salas-Romero et al., 2015) can be used for determining the presence of potential quick clays, coarse-grained materials or top of bedrock.

Although the correlation between different geophysical data sets seems to work well (Fig. 5, 6, 7 and 8), this study shows that the reflection seismic method allows higher resolution delineation of the bedrock surface and coarse-grained layer compared to the other methods. Electromagnetic methods (e.g. RMT) help to discriminate clay from sand deposits as well as leached from unleached clays (if the shallow clay layer is not too thick). Thus, they complement the reflection seismic data when studying areas prone to quick-clay landslides. Borehole data at the site (BGA, 2018) show that the coarse-grained layer can be found in many points to the north and south of the study area, although in some cases it is only a very thin layer (0.5 to 1 m). This wide extension is consistent with the reflection seismic results since the coarse-grained layer can be delineated in most of the lines (Fig. 5, 6, 8, see also Malehmir et al., 2013b).

Fractured or disturbed materials are identified in all the seismic lines. The six-channel river seismic data show several fracture zones. These fracture zones, as well as some of the fractured materials interpreted in the land seismic lines coincide with the position of morphological or geological lineaments (usually fracture zones) shown in the geological map (Fig. 2a). Using complimentary geophysical methods helps interpreting these structures. For example in the case of line 5–5b (Fig. 6b), the reflection seismic data do not show any fault at the river position, however the difference in time (1 s) between the





cabled geophone and wireless data already indicated a possible fracture in the river. In comparison, ATEM resistivity results (Bastani et al. 2017; Malehmir et al., 2016) show a low resistivity zone at the river position (Fig. 6e), which is interpreted as a possible fault. A fracture zone is also identified in the six-channel river seismic data at the same position (Fig. 10b and 11), as a zone of lost reflectivity. It is interesting to notice that the interpreted fracture system occurs where a quick-clay landslide

scar is present. We interpret the combination of coarse-grained layer, bedrock morphology and presence of fractured bedrock to contribute to the formation of quick clay and to be likely pre-conditioning factors for landslide (see e.g. L'Heureux et al., 2017).

**5.2 3D Modelling of the subsurface materials**

One of the main objectives of this study was to obtain the extension of the coarse-grained layer and its spatial relationship

with the bedrock surface. Malehmir et al. (2013a, 2013b) have shown that the coarse-grained layer extended locally in a restricted area, but this work shows the extension of this layer to the north and south of the initial study area. The coarse-grained and bedrock horizons were picked on the processed land and river seismic lines, and elevation surfaces were interpolated using the seismic, borehole (BGA, 2018) and LiDAR data (© Lantmäteriet). Total sounding data identify top and bottom of the coarse-grained at many points at the site. The LiDAR data were mainly used for fixing the elevation of the

rock outcrops. The surfaces for the top of the coarse-grained layer and top of bedrock were calculated using a natural neighbour interpolation after a Delaunay triangulation of the scattered sample points was generated. Figure 12 shows the results of this modelling together with the 3D visualization of the seismic profiles. The bedrock surface in Fig. 12c undulates between 80.5 and -110 m elevation. Two elongated depressions next to the river, which cross lines 2–2b, 5–5b, 6 and 7, can be identified. These depressions are interpreted as possible faults and coincide with the position of fracture zones (Fig. 2a).

The gentler coarse-grained layer surface in Fig. 12d ranges from 18.5 to -28.5 m elevation. The data are better constrained in the area surrounding the rock outcrop, as more seismic lines are available there. The bedrock surface is visible in all the lines, except line 4 (Malehmir et al., 2013b) and, therefore, the model of the bedrock surface in the study area is generally well constrained. The coarse-grained layer is not identified in the northwestern part of line 2–2b and line 6, but the model gives a good overview of the layer extension, which spreads over both sides of the river, and is an important feature in

studying quick-clay landslides.

The maximum elevation for both the bedrock and coarse-grained layer surfaces coincides with the centre of the survey site, and the undulated bedrock dips down towards the river. This implies that more water may flow in the coarse-grained layer closer to the river. Löfroth et al. (2011) and Salas-Romero et al. (2015) show that the coarse-grained layer is thicker within the depressions, which is related to the deeper top of bedrock and sediment focusing when deposition took place. The

thickness of the coarse-grained layer and the higher water flow could increase the thickness of potential quick clays in those depressions. The possible faults indicate areas more susceptible to slide, due to slope inclination and/or increased water flow. Figure 13 shows a view of the 3D modelling from the north. Figure 13c shows the interpolated elevation surface for top of bedrock along the river. Note the bedrock undulation between lines 6 and 7, whose reflection seismic results (Fig. 7 and 8)





indicate 3D effects in the western part of the profiles due to rough topography. Figure 13d shows the delineation of the coarse-grained layer along the river; the reflections corresponding to the filled channels correlate with the interpolated top of the coarse-grained layer surface (see Fig. 13b and 13d).

### 5.3 3D Hydrological modelling of the coarse-grained layer

After obtaining the elevation surface for the top of the coarse-grained layer, we modelled the elevation surface for the bottom of the layer using the RMT resistivity (Lindgren, 2014; Shan et al., 2014; Wang et al., 2016) and available borehole data (BGA, 2018; Salas-Romero et al., 2015). Thickness values of the coarse-grained layer were picked along the RMT resistivity profiles at its estimated position. The thickness of the layer was then interpolated together with thickness values obtained from the borehole data. The interpolated thickness surface was subtracted from the elevation surface of the top of the coarse-

grained layer previously modelled in order to obtain the elevation surface of the bottom of the coarse-grained layer.

The elevation and thickness values for the coarse-grained layer were used for obtaining a single-layer two-dimensional groundwater model (Fig. 14a), which reflects a gradual decrease of the hydraulic head from the outcrop area to the river. The horizontal model resolution is 10 m. Groundwater table from the existing boreholes in the study area (see Fig. 2b and 14a) matches the groundwater level obtained in the model. Boreholes 7056, 7060 and 7069 do not fit the groundwater model,

possibly due to having only one measurement of the water table, which may have fluctuated at a later stage. The total volume (V) of the interpolated coarse-grained layer was estimated around 5 hm³. The steady-state confined groundwater flow Eq. (1) (Bear, 1972) is:

$$div\left(T.grad(h)\right) = 0, \tag{1}$$

where the hydraulic head (h) is the unknown variable, was solved using a finite volume method (Guyer et al., 2009). The

transmissivity (T) was defined as the product of the local interpolated coarse-grained layer thickness with a uniform hydraulic conductivity (K), the latter needed to be calibrated (Fig. 14b). The transmissivity values are higher (from 0.006 to 0.016 m²/s) around the land seismic lines, due to a thicker coarse-grained layer at these positions. As hydraulic boundary conditions, the cells directly underlying the river were assumed to have a fixed head corresponding to the average river level as shown in Fig. 2b. The central part of the model is characterized by the disappearance of the coarse-grained layer, a thin

sandy-silty till cover and several elevated rock outcrops (Fig. 2a), which we hypothesized to be a groundwater recharge area for the coarse-grained layer by infiltration along the bedrock/sediments interface. The area of the considered recharge zone (A) is approximately 0.2 km² and the groundwater recharge rate was used as a calibration parameter. The recharge area was represented as a layer of constant T, equal to 0.001 m²/s, in order to distribute the recharge to the adjacent coarse-grained layer. After a few tests, the groundwater recharge rate over the rest of the model domain was deemed unlikely to exceed 5 %

of the normal annual mean net precipitation (Pnet) in the catchment estimated at around 460 mm/year (Swedish Meteorological and Hydrological Institute–SMHI, 2018), and was effectively set to zero.

Groundwater level measurements, although sparse, indicate either large seasonal variations of the water table or sporadic connections to a deeper groundwater system via bedrock fracture zones. In the present model, the coarse-grained layer



hydraulic conductivity and the recharge rate over the central outcrops were manually fitted against the groundwater head measurements in the available boreholes (measurement dates are April/May of 2010 and middle March of 2013). The model predicts an effective K of 0.001 m/s, which is plausible for a coarse sandy layer, and a recharge rate over the central part of the model of about 210 mm/year, which is about 45 % of the normal net precipitation for this catchment (SMHI, 2018).

According to this model, the groundwater recharge through the top fine-grained sediment layers is of second order to explain the hydraulic behaviour of the coarse-grained layer. Artesian conditions are not found to be prevalent under 'spring' base conditions but a shift to seasonal artesian conditions during the high-water season (from June to December) cannot be ruled out in light of the available data. A calculation of the water residence time (tau) in the coarse-grained layer can be done according to Eq. (2):

$$tau = \frac{V*phi}{A*Pnet*0.45},$$                                                                            (2)

which is approximately 20 years, assuming a porosity (phi) of 0.2 (or 20 %). This relatively short residence time would point to lower salinity groundwater occurrences in the coarse-grained layer compared to the overlying clays and ion transfer from the clays to the underlying groundwater flow system, either by diffusion or by slow infiltration (Torrance, 1979), which has been identified as a precursor to the formation of quick clays (Rankka et al., 2004).

Figure 14c shows the values of the mean groundwater velocity (Darcy flow vector amplitude divided by phi) and vector field of the Darcy flow, whose directions go from the outcrop area to the river. At positions where T (or thickness of the coarse-grained layer) is lower (less than 0.006 m²/s), the mean groundwater velocity is higher (from 0.00005 to 0.00015 m/s), and vice-versa, where T is large (from 0.006 to 0.0162 m²/s) the velocity is very low (less than 0.00012 m/s). Assuming that the leaching of marine salts increases with groundwater velocity, areas where the groundwater velocity is higher could be at an

increased risk of quick-clay formation.

**5.4 Morphology of the Göta River valley**

The total-field magnetic data were corrected for differences in diurnal variations and instrumental drift using the base station data (a background, International Geomagnetic Reference Field–IGRF, value of 50600 nT, Earth magnetic field, was subtracted from the results to convert to residual magnetic anomaly data). However, several inconsistencies were still present

in the corrected data such as different values at overlapping positions, level jumps between measurement days (see sketch with the measurement and base station positions in the lower right corner in Fig. 15), and elongated features parallel to the measuring paths. In order to level the data, a constant value was added or removed in the measurements for the last three days, and the polarity was changed for the data of the first day. For removing the elongated features, the data were divided in two groups, the first three days and the last two days of measurements, gridded and filtered similar to the micro-levelling

procedure (Minty, 1991). A low-pass filter was applied in N-S direction, a high-pass filter in the E-W direction, and then the result was subtracted from the original gridded data. The final result (Fig. 15) is smoother and more homogeneous than the



initial data but still has few elongated features, which are residual errors (acquisition footprint) and not natural features as they coincide with the sampling directions.

Figure 15 shows that the residual magnetic anomaly values are higher on the northern part than on the southern (ranging from -20 to 20 nT); at the bottom and in the eastern flank of the landslide scar that crosses line 5–5b the values reach 20 nT. Almost all along line 4 and between lines 2–2b and 5–5b higher residual magnetic anomaly values are found. Salas-Romero et al. (2015) showed that the coarse-grained layer has higher magnetic susceptibility compared to the sediments above and below of these materials. Besides, we also took samples at the bottom of the landslide scar up to 1.5-2 m depth that were identified as silty-sandy. The high residual magnetic anomaly values at the landslide scar may indicate that the coarse-grained layer, which drains the water infiltrated from the nearby outcrops and/or fractures, may have acted as sliding surface together with quick clays. L'Heureux et al. (2012) identified a 'weak layer' composed of softer and more sensitive clays and sands (compared to the surrounding materials) that acted as slide prone layer initiating the 1996 Finneidfjord landslide. This layer also contained biogenic gas, which may have affected its geotechnical properties. Biogenic gas was found in boreholes BH2 (Salas-Romero et al., 2015) and 7075 (BGA, 2018), which adds another similarity with the case described in L'Heureux et al. (2012). The landslide scar was formed approximately 50 years ago according to the farmer working these lands, and the landslide could have been triggered due to changes in the river water level and/or an increase in pore pressure, and/or toe erosion. Looking the borehole information available in the area (BGA, 2018; Löfroth et al., 2011; Salas-Romero et al., 2015), the top of the coarse-grained layer lies between 10 and 30 m depth, being deeper at boreholes 7206 and BH2. At the latter borehole, the thickness of the coarse-grained layer reaches almost 10 m (in BH1 and BH3 the thickness of this layer is around 1 and 3 m, respectively). The shallowest tops of the coarse-grained layer are registered at borehole 7202 at the landslide scar and at BH1. Comparing this information with the residual magnetic anomaly data, we infer that the coarse-grained layer, its distance to the surface and thickness, may be related to the high values of the magnetic data in the northern part and around BH1 where line 1 lies. The bedrock at these positions is quite deep to cause such high values (e.g. in BH2 the top of bedrock is around 78 m). The southwestern part does not reflect the same behaviour, which may be related to the thinness of the coarse-grained layer and/or its depth. On the southern part of the residual magnetic anomaly map an elongated anomaly that follows SW-NE direction is observed. The values are higher in the centre (ranging approximately between -15 and 5 nT), with the maximum values around borehole 7067 (BGA, 2018) in the intersection with line 2–2b, and lower around the anomaly (ranging from -25 to -15 nT). At borehole 7067, bedrock is close to the surface, with high residual magnetic anomaly values coinciding with the bedrock topography. We interpret the elongated anomaly to be related to the scarp formed between bedrock and the lower elevated sediments to the west (Fig. 2). The southeastern part of the residual magnetic anomaly map where there are several outcrops and the surface elevation is higher, includes negative and positive magnetic anomalies. A few houses and farms may have influenced the residual magnetic anomaly data in this area.

Figure 16 presents three examples of combining the side-scan sonar (© SGU) with the bathymetric data (© SGI) on the river (see positions in Fig. 2c). The first example, Fig. 16a–e, is a section that is crossed by line 5–5b. Deposits of unclear origin can be observed at the bottom of the section (Fig. 16a); their granular texture and darker colour indicate that they may be



harder and denser than the surrounding materials. These deposits may proceed from the landslide scars in both sides of the river (Fig. 2) and/or from the discharge of sediments transported by the gullies or tributary intersecting the Göta River at this position. Slopes in profile AA' can be classified as "high terrace-steep slope" (Millet, 2011), with values ranging between 30° and 45°. Profile BB' (Fig. 16d–e) shows the same type of slope on the northern riverbank, and a gentler slope (the slope

is a mix between "high terrace-steep slope" and "straight un-even profile", Millet, 2011) on the opposite side. The B' extreme coincides with the position of a stream (Fig. 2). The toe of slope probably contains material due to the collapse of the subaquatic slope or sediments deposited by the stream. The second example, Fig. 16f–j, is a section located south of the study area. Profile AA' shows a possible subaquatic landslide scar in the western side. Figure 16g–h shows a "double terrace" (slopes between 40° and 55°) on the western side and a "high terrace-steep slope" (inclination around 47°) on the

opposite side. The deepest terrace in the western side has a ledge that is around 5 m high, which coincides with the position of the identified subaquatic landslide scar. Profile BB' (Figure 16i–j) shows accumulated material against the riverbank in the eastern side. The slope on the western side can be classified as a "high terrace-steep slope" (slope ~50°), and the slope profile on the opposite side resembles a combination of the classes "high terrace-steep slope" and "straight un-even profile" (maximum slope is around 30°). The toe of slope seems to consist of landslide deposits. The third example, Fig. 16k–m, is a

section even more to the south than the one in Fig. 16f–j. Accumulated material is visible at the bottom of the river in the eastern side in profile AA' (Figure 16l). The inclination of the slope is very irregular, generally below 20° but with some parts having higher inclination. The slope looks like a "straight un-even profile", although there are parts with small terraces. The toe of slope appears to have formed from landslide deposits. The opposite side resembles a "high terrace-steep slope" (slope is 45°).

Erosion and landslide processes have formed the landscape of the Göta River valley (SGI, 2012a). Land and river seismic data, together with side-scan sonar and bathymetric data give a good overview of the (sub-)surface in this valley. Along the river filled channels are identified, which were probably formed when the riverbed morphology changed over time. The bedrock at the river channel shows several fracture zones. The slopes of the riverbanks are generally steep and many subaquatic landslide scars and deposits can be found along the river. The origin of these deposits may also be related with

the remains from the erosion protections placed between 1960 and 1970 along the Göta River. The likelihood of retrogressive landslides inland can increase due to undercutting of riverbanks during high discharge or wave erosion generated by shipping movement, which reduces lateral support and causes more instability (SGI, 2012a). At the surface, the inclination of the slopes is influenced by the land use and precipitation induced processes (SGI, 2012a).

According to SGI, our study area has, in general, medium landslide risk. Close to the river the risk is higher due to the

presence of highly sensitive clays or quick clays. SGI has evaluated the risk of landslides along the Göta River under different climate change scenarios (2012a, 2012c). These scenarios estimate increases in temperature by 4-5° and 20-30 % more precipitation by 2100. High and low drainage levels from Lake Vänern will be more frequent, sea level will rise up to 0.7 m at Göteborg, and maximum groundwater levels and pore pressure in slopes will not change significantly. These conditions could increase the occurrence of landslides and modify the valley morphology in the future (SGI, 2012a). Erosion




has a great impact to the south of Lilla Edet, and under these climate change scenarios will be more intense along the river's course (SGI, 2012a). This means that although the study area is located to the north of Lilla Edet, where erosion has less impact, any small change in the erosion rate in the future could cause serious effects as this area is already considered to be at medium-high landslide risk.

**6 Conclusions**

Through an extensive reflection seismic investigation, which includes land and river lines and their combination with other geophysical, geotechnical and borehole data, this study allows the delineation in a regional scale of a coarse-grained layer, underlying bedrock and fracture zones in an area prone to quick-clay landslides in southwest Sweden. This is the first time in Sweden that land and river reflection seismic data are combined for studying the subsurface associated with this type of

landslides.

Some of the geological and geohydrological prerequisites for the formation of quick clay in nature specified by Rankka et al. (2004), are shown in the results of this study. Correlation of reflection seismic, resistivity and P-wave refraction tomography results offers information about the presence of underlying coarse-grained layers, peaks in the bedrock surface, and approximated thickness and type of clay deposits. 3D modelling of the elevation surfaces for the coarse-grained layer and

bedrock illustrates the possible infiltration points in the area, as nearby elevated rock outcrops or fractures. Hydrological modelling of the coarse-grained layer estimates the size of the catchment area and the dominant leaching processes for the low-water season (diffusion and slow infiltration). The formation of quick clays is more significant under artesian groundwater conditions, which may be dominant in the high-water season (summer and autumn) at the survey site. Ground magnetic data delineate coarse-grained materials and bedrock topography. The northern part of the study area contains a

shallower and thicker coarse-grained layer. At the bottleneck landslide scar present in the centre of the survey, the high residual magnetic anomaly values indicate the presence of the coarse-grained layer, which may have acted as sliding surface together with quick clays. The side-scan sonar and bathymetric data reveal a number of distinct morphological features in the Göta River valley, that reflect the erosional processes, landslide scars and mass-movement deposits.

This work illustrates the significance of studying subsurface geology, including features within bedrock that are often

overlooked when investigating landslides, especially ones that involve quick clays.

*Data availability*. The land reflection seismic results and modelling, magnetic results and hydrological modelling are available online with restricted access (permission from Alireza Malehmir is necessary before the data can be accessed) at https://doi.org/10.5878/acbv-h350 (Salas-Romero et al., 2019a), https://doi.org/10.5878/md19-qw71 (Salas-Romero et al., 2019b), https://doi.org/10.5878/a8xn-fc97 (Salas-Romero et al., 2019c), https://doi.org/10.5878/19t9-js25 (Salas-Romero et

al., 2019d), respectively. The river reflection seismic, side-scan sonar and geological raw data are available from SGU following registration. The bathymetric raw data are available from SGI following registration, and the geotechnical data are



available online at http://bga.swedgeo.se/bga/. The resistivity and tomography modelling results are properly cited and referred to in the reference list.

*Author contributions.* SS and AA participated in the acquisition of the land reflection seismic and magnetic data. AA designed the fieldwork campaigns in 2011 and 2013 in the study area. SS processed the land and river reflection seismic data

with the support from AA. SS performed the geological modelling. SS analysed, corrected and processed the magnetic data. SS prepared the data used for the hydrological modelling, which was performed by BD. BD wrote most of the hydrological modelling section. SS made an integrated interpretation of the geophysical, geotechnical, geological and hydrological data with the help of all co-authors. SS is the main contributor to the writing of this article. All co-authors contributed to the final version of this article.

*Competing interests.* The authors declare that they have no conflict of interest.

*Acknowledgements.* The GWB program of SEG and Uppsala University sponsored this project. The geophysical data were collected with the help of PhD and MSc students from Uppsala University, and staff from SGU, in particular S. Ohlsson. This study was initiated as part of a joint research collaboration among Uppsala University, SGU, Leibniz Institute for Applied Geophysics, University of Cologne, Syiah Kuala University, Polish Academy of Sciences, Norwegian Seismic

Array, Norwegian Geotechnical Institute, Institute for Geosciences at the University of Oslo, Geotechnical Group at the Norwegian University of Science and Technology, and Geological Survey of Norway. Partial funding from Trust2.2-Geoinfra project (http://trust-geoinfra.se/) supported this work (252-2012-1907). We are thankful to SGU, especially to B. Bergman, and SGI for providing data. We would like to thank S. Wang, C. Shan, A. Lindgren and M. Bastani for providing their resistivity and tomography modelling results. S. Andersson helped in the processing of one of the land seismic lines as

part of her MSc studies. GLOBE Claritas™ under license from GNS Science, Lower Hutt, New Zealand was used for processing the seismic data. Figures were prepared using Generic Mapping Tools (http://gmt.soest.hawaii.edu/), Inkscape (https://inkscape.org/), and Paradigm GOCAD®. Side-scan sonar, bathymetric, and magnetic data were processed and represented using MATLAB®. OpendTect (https://www.opendtect.org) and MATLAB® were used for obtaining the seismic horizons and interpolated elevation surfaces between the seismic sections. FiPy (https://www.ctcms.nist.gov/fipy) and

MATLAB® were used for the hydrological modelling.

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

**Table 1.** Main acquisition parameters of the seismic lines (March 2013).



| | Line 2b | Line 5b | Line 6 | Line 7 |
|---|---|---|---|---|
| *Survey parameters* | | | | |
| Acquisition System | Sercel 428 ™ | Sercel 428 ™ | Sercel 428 ™ | Sercel 428 ™ |
| Number of Receivers | 160 | 324 cabled geophones, 50-1C and 24-3C MEMs planted wireless recorders | 133 | 100 |
| Number of Shots | 157 | 87 | 129 | 100 |
| Shot/ Receiver Spacing | 4 m | 20 m/4 m (10 m for wireless recorders) | 4 m | 4 m |
| Maximum Offset | 636 m | ~2300 m | 528 m | 396 m |
| Source Type | Accelerated weight drop | Dynamite (50 to 200 g) | Accelerated weight drop, sledgehammer, dynamite (50 g) | Sledgehammer, dynamite (50 g) |
| *Recording parameters* | | | | |
| Record Length | 10 s | 10 s | 10 s | 10 s |
| Sampling Rate | 0.5 ms | 0.5 ms/1 ms (wireless recorders) | 0.5 ms | 0.5 ms |
| *Receiver and source parameters* | | | | |
| Geophone Frequency | 28 Hz | 28 Hz, 1C-10Hz and 3C-MEMs | 28 Hz | 28 Hz |
| Number of Geophones per Set | Single | Single | Single | Single |




| Source Pattern | 3 to 5 impacts/point | Single point/hole | 5 impacts/point, single point/hole | 3 to 5 impacts/point, single point/hole |
|---|---|---|---|---|
| Shot Depth | 0 m | 0.8 to 1 m | 0 m, 0.9 m | 0 m, 0.9 m |

**Table 2.** Lateral (L) and vertical (V) resolutions of each method.

| Remote sensing | | Near surface geophysics | | Boreholes | |
|---|---|---|---|---|---|
| Geology | L: N/A<br>V: N/A | Land reflection seismic | L: 27-35 m<br>V: 4-7 m | Natural gamma log | L: N/A<br>V: 0.01 m |
| LiDAR | L: 2 m<br>V: 2 m | River reflection seismic | L: 4-17 m<br>V: 0.1-2 m | Magnetic susceptibility log | L: N/A<br>V: 0.25-0.5 m |
| Side-scan sonar | L: 20 cm<br>V: N/A | Ground magnetics | L: 0.4 m<br>V: N/A | Total sounding | L: N/A<br>V: 0.025 m |
| Bathymetry | L: 1 m<br>V: 1 m | | | | |

5  **Table 3.** Main processing steps applied to both land and river seismic data.



| Step | Parameters |
| --- | --- |
| *Land reflection seismic data* | |
| 1 | Read 0.5 s SEG-Y data |
| 2 | Zero-time correction |
| 3 | Vertical shot stacking (lines 2b, 6 and 7) |
| 4 | Merge lines from 2011 and 2013 (lines 2 and 2b, and lines 5 and 5b) |
| 5 | Extract and apply geometry: CDP spacing 2 m and 10 m (line 5-wireless) |
| 6 | Trace editing |
| 7 | First break picking: automatic picking and manually corrected |
| 8 | Elevation static corrections: datum 20-25 m and replacement velocity 1500 m/s |
| 9 | *fk*-filter (line 5-wireless) |
| 10 | Wiener deconvolution: gap 10-15 ms |
| 11 | Band-pass filter: 40-60-200-250 Hz (sledgehammer), 40-60-180-200 Hz (weight drop), 70-80-260-280 Hz (dynamite-cabled) and 30-60-180-200 Hz (dynamite-wireless) |
| 12 | Spectral whitening: 70-80-180-200 Hz (dynamite-cabled) and 70-80-200-220 Hz (dynamite-wireless) |
| 13 | Top mute using first breaks |
| 14 | Surgical mute (line 5-wireless) |
| 15 | Airwave mute |
| 16 | AGC: 70-500 ms |
| 17 | Velocity analysis |
| 18 | Residual static corrections (lines 6 and 7) |





| 19 | Normal moveout (NMO) corrections: 50-150% stretch mute |
|----|----|
| 20 | Stack |
| 21 | Band-pass filter: 20-40-150-170 Hz (sledgehammer), 50-60-150-170 Hz (weight drop) and 30-40-110-130 Hz (dynamite-wireless) |
| 22 | Post-stack deconvolution (line 5-wireless) |
| 23 | *fx*-deconvolution |
| 24 | Trace balance |
| 25 | *fk*-filter (lines 6 and 7) |
| 26 | Time-to-depth conversion using constant velocity of 1500 m/s |

*River reflection seismic data*

| 1 | Read 0.2 s SEG-Y data |
|----|----|
| 2 | Apply marine geometry |
| 3 | Wiener deconvolution: gap 2-6 ms |
| 4 | AGC: 100 ms |
| 5 | NMO corrections: 30% stretch mute |
| 6 | Stack |
| 7 | Post-stack deconvolution |
| 8 | Top mute |
| 6 | *fx*-deconvolution |
| 7 | Trace balance |
| 8 | Time-to-depth conversion using constant velocity of 1500 m/s |





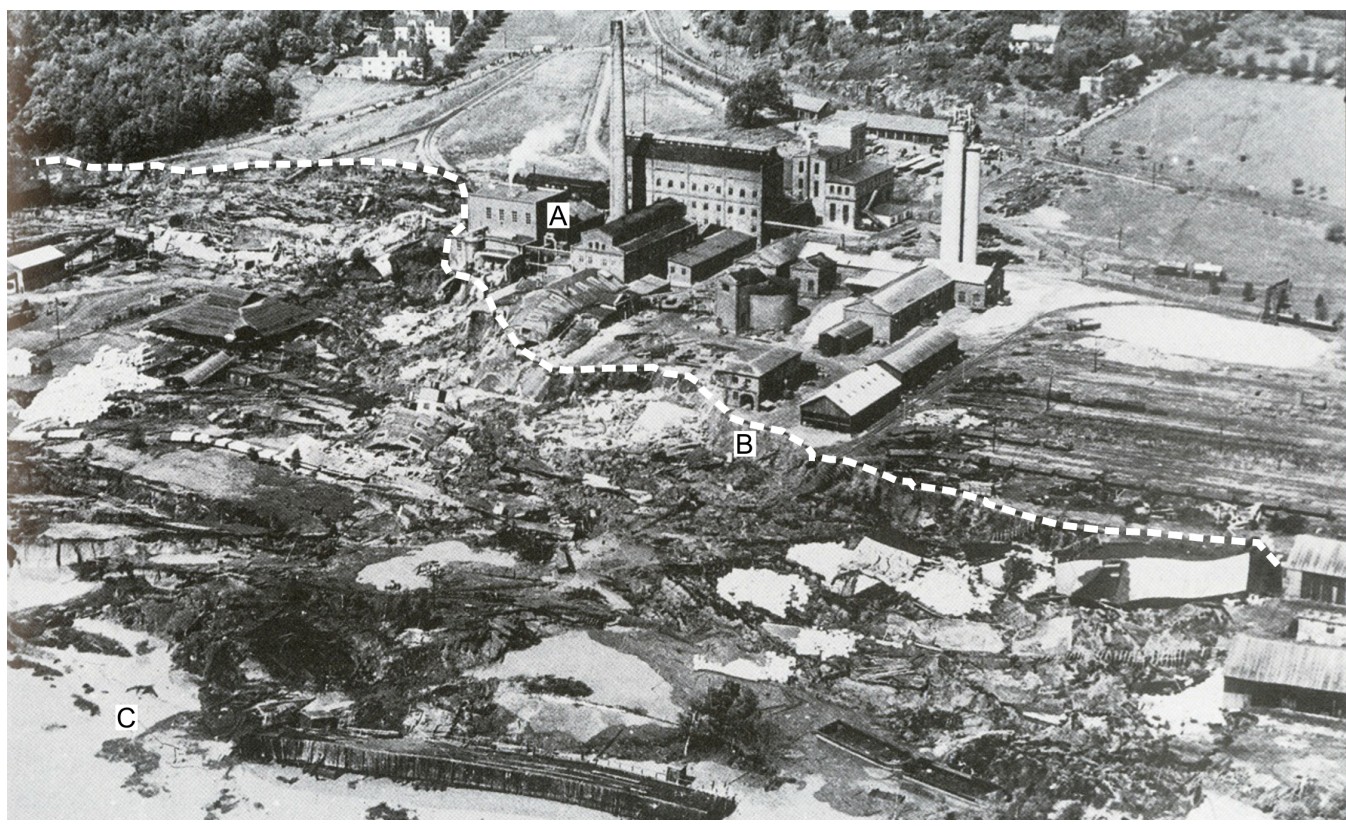

**Figure 1.** Aerial photo from the 1957 Göta landslide. The picture shows the sulfite factory located next to the river where the landslide started. A: factory, B: slide back wall and C: river. Photo: Edet group's archive.





**Figure 2.** Quick-clay study site location. **(a)** Geological map of the study area (© SGU), Fråstad, in southwest Sweden. The legend on the right of the map shows the geological materials. The legend below the map shows the symbols for different geological structures and features, as well as some of the investigations carried out by Uppsala University, SGU and SGI. Shot positions SH5 and SH7 (black and white stars) from line 5–5b and line 7, respectively. **(b)** LiDAR elevation map of the study area (© Lantmäteriet). SGI boreholes are represented as black and white circles, and Uppsala University boreholes as blue circles. Highway E45 is visible as a straight line in the eastern part of the map. **(c)** Sketch of the Göta River that includes the position of the study area (Fig. 2b), and the position of figures shown in Section 5.4. The red arrows indicate the beginning and end of the river seismic lines.

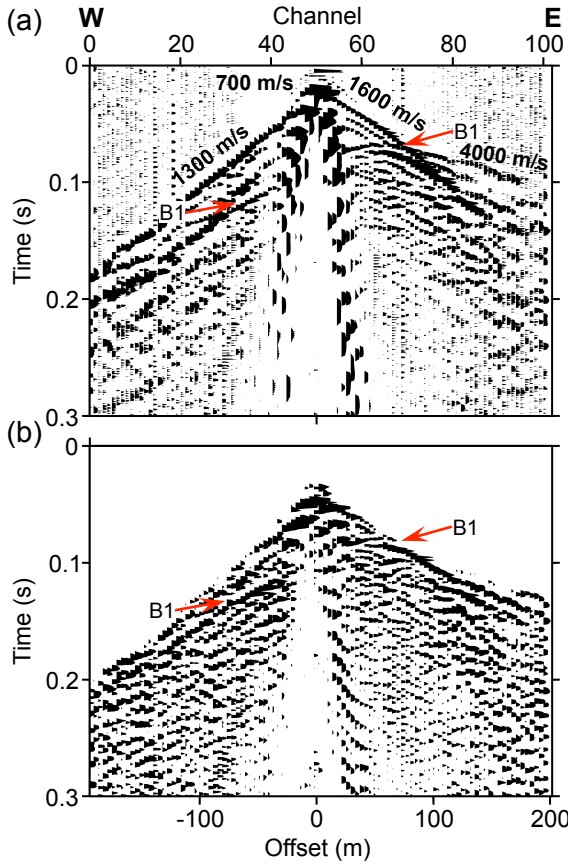

**Figure 3.** Shot record along line 7 (see SH7 in Fig. 2a). **(a)** Raw shot gather (sledgehammer was used as seismic source) with trace balance applied. Apparent velocities for different types of arrivals are shown at the top of each event. **(b)** A reflection becomes clearer after a series of pre-stack processing steps: elevation static corrections, Wiener deconvolution, band-pass filter, trace editing, removal of first breaks and trace balance. Scale 3H:4V.



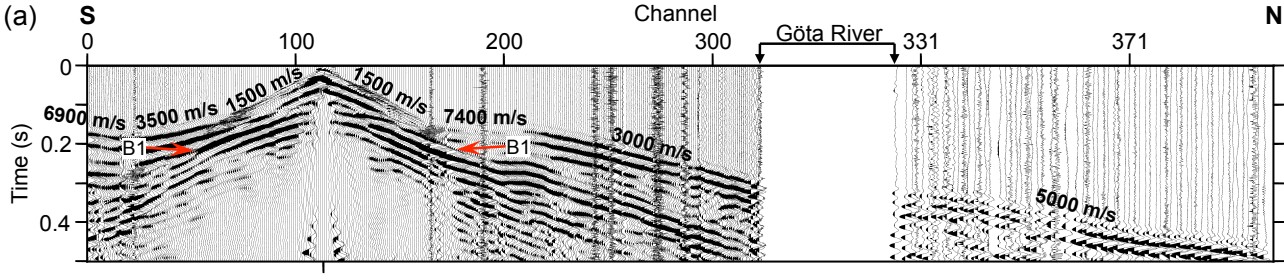

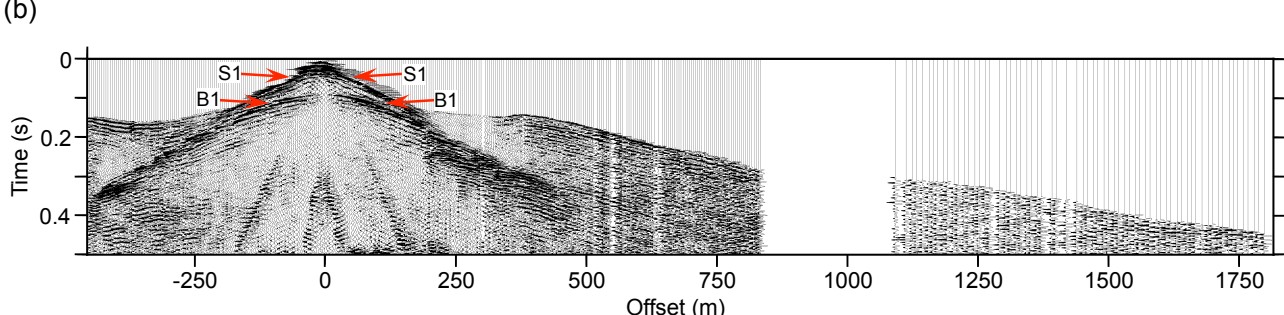

**Figure 4.** Shot record along line 5–5b (see SH5 in Fig. 2a). **(a)** Raw shot gather (dynamite, 50 g, was used as seismic source). Apparent velocities for different types of arrivals are shown at the top of each event. **(b)** Two sets of reflections seem to be revealed, B1 (bedrock) and S1 (coarse-grained materials), after a series of pre-stack processing steps: elevation static corrections, band-pass filter, spectral whitening, trace editing, removal of first breaks and AGC.



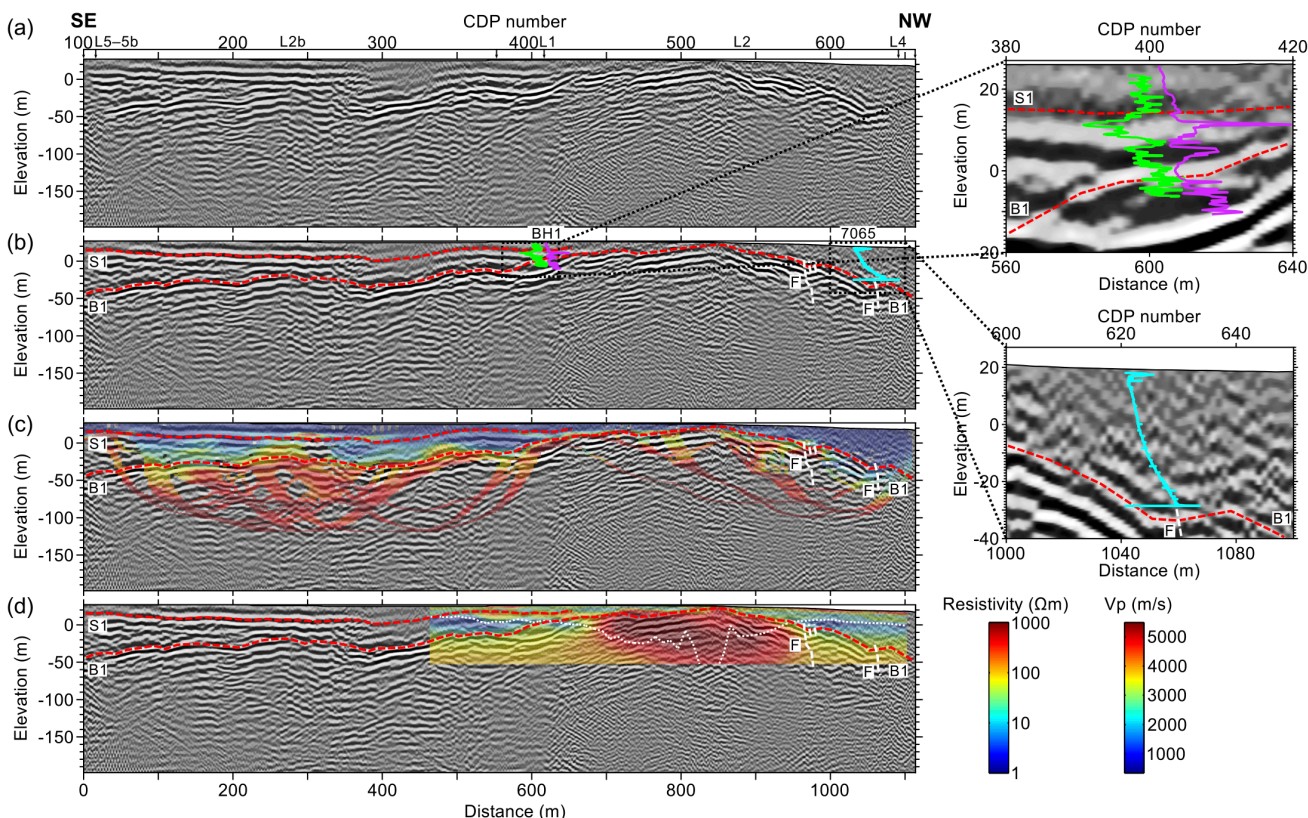

**Figure 5.** Merged land seismic line 2–2b. **(a)** Reflection seismic processing results. On top of the section the corresponding parts of each line and the position of the lines that intersect the merged line are indicated. **(b)** Interpreted seismic section that includes three borehole data sets, natural gamma radiation (Salas-Romero et al., 2015) from BH1 in green (ranging from 87 to 182 API, distance to the seismic line 13.4 m), magnetic susceptibility (Salas-Romero et al., 2015) from BH1 in purple (ranging from $0.09 \cdot 10^{-6}$ to $2.2 \cdot 10^{-6}$ m$^3$/kg), and total sounding (BGA, 2018) from borehole 7065 in blue (ranging from 0 to 15 kN, distance to the seismic line 0.02 m). Close-ups for each borehole are shown on the upper right side of the figure. S1: coarse-grained materials and B1: bedrock represented by dashed red lines, and F: fractured or disturbed materials represented by a thick dashed white line. **(c)** P-wave refraction tomography model (Wang et al., 2016) superimposed on the interpreted seismic section. **(d)** RMT resistivity model (Wang et al., 2016) superimposed on the interpreted seismic section. The thin dashed white line indicates the depth above which the results are considered reliable. Resistivity and P-wave velocity values are shown in the columns located on the bottom right side of the figure.

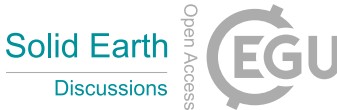


**Figure 6.** Merged land seismic line 5–5b. On top of the section the corresponding part of line 5 from 2011, the position of the lines that intersect the merged line, the landslide scar and Göta River are indicated. **(a)** Reflection seismic processing results. The fold distribution along line 5–5b is represented by a blue line at the bottom part of the section (ranging from 0 to 115). In the cabled geophone part (south) the maximum value is 115 and in the wireless part (north) 60. Note the lower fold between CDPs 400 and 480 (minimum value is 47). **(b)** Interpreted seismic section that includes three borehole data sets, natural gamma radiation (Salas-Romero et al., 2015) from BH3 in green (ranging from 59 to 200 API, distance to the seismic line 0.23 m), magnetic susceptibility (Salas-Romero et al., 2015) from BH3 in purple (ranging from $0.07 \cdot 10^{-6}$ to $2.7 \cdot 10^{-6}$ m³/kg), and total sounding (BGA, 2018) from borehole 7062 (ranging from 0 to 12 kN, distance to the seismic line 90.3 m) in blue. Close-ups for each borehole are shown at the bottom part of the figure. S1: coarse-grained materials and B1: bedrock represented by dashed red lines, and F: fractured or disturbed materials represented by a thick dashed white line. **(c)** P-wave refraction tomography model (Wang et al., 2016) superimposed on the interpreted seismic section. P-wave velocity values are shown on the right side of the figure. **(d)** RMT resistivity results (Wang et al., 2016) superimposed on the interpreted seismic section. The thin dashed white line indicates the depth above which the results are considered reliable. Resistivity values are shown on the right side of the figure. **(e)** ATEM resistivity results (Bastani et al., 2017) superimposed on the interpreted seismic section. Resistivity values are shown on the right side of the figure.

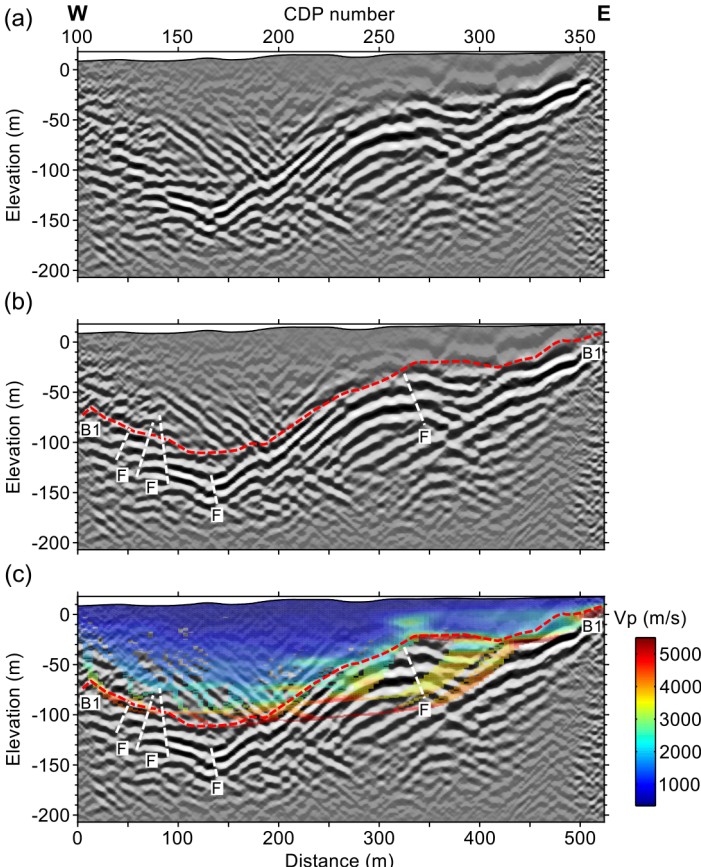

**Figure 7.** Land seismic line 6. **(a)** Reflection seismic processing results. **(b)** Interpreted seismic section, with only a major reflection, B1, identified from bedrock (dashed red line). At different positions, several fractured or disturbed materials (F) are also delineated using a





dashed white line. **(c)** P-wave refraction tomography model (Wang et al., 2016) superimposed on the interpreted seismic section illustrating consistency between the two independent methods. P-wave velocity values are shown on the right side of the figure.

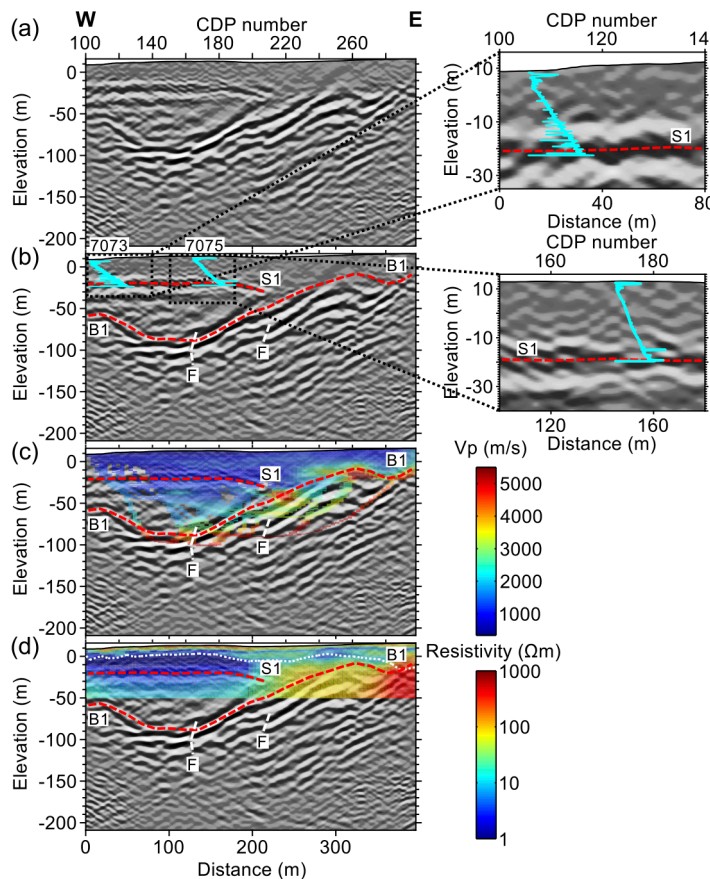

**Figure 8.** Land seismic line 7. **(a)** Reflection seismic processing results. **(b)** Interpreted seismic section including total sounding data (BGA, 2018) from borehole 7073 (ranging from 0 to 14 kN, distance to the seismic line 29.3 m) and 7075 (ranging from 0 to 14 kN, distance to the seismic line 1.6 m), both in blue. Close-ups for each borehole are shown on the upper right side of the figure. S1: coarse-grained materials and B1: bedrock represented by dashed red lines, and F: fractured or disturbed materials represented by a thick dashed white line. **(c)** P-wave refraction tomography model (Wang et al., 2016) superimposed on the interpreted seismic section. P-wave velocity values are shown on the right side of the figure. **(d)** RMT resistivity results (Wang et al., 2016) superimposed on the interpreted seismic section. The dashed white line indicates the depth above which the results are considered reliable. Resistivity values are shown on the right side of the figure.





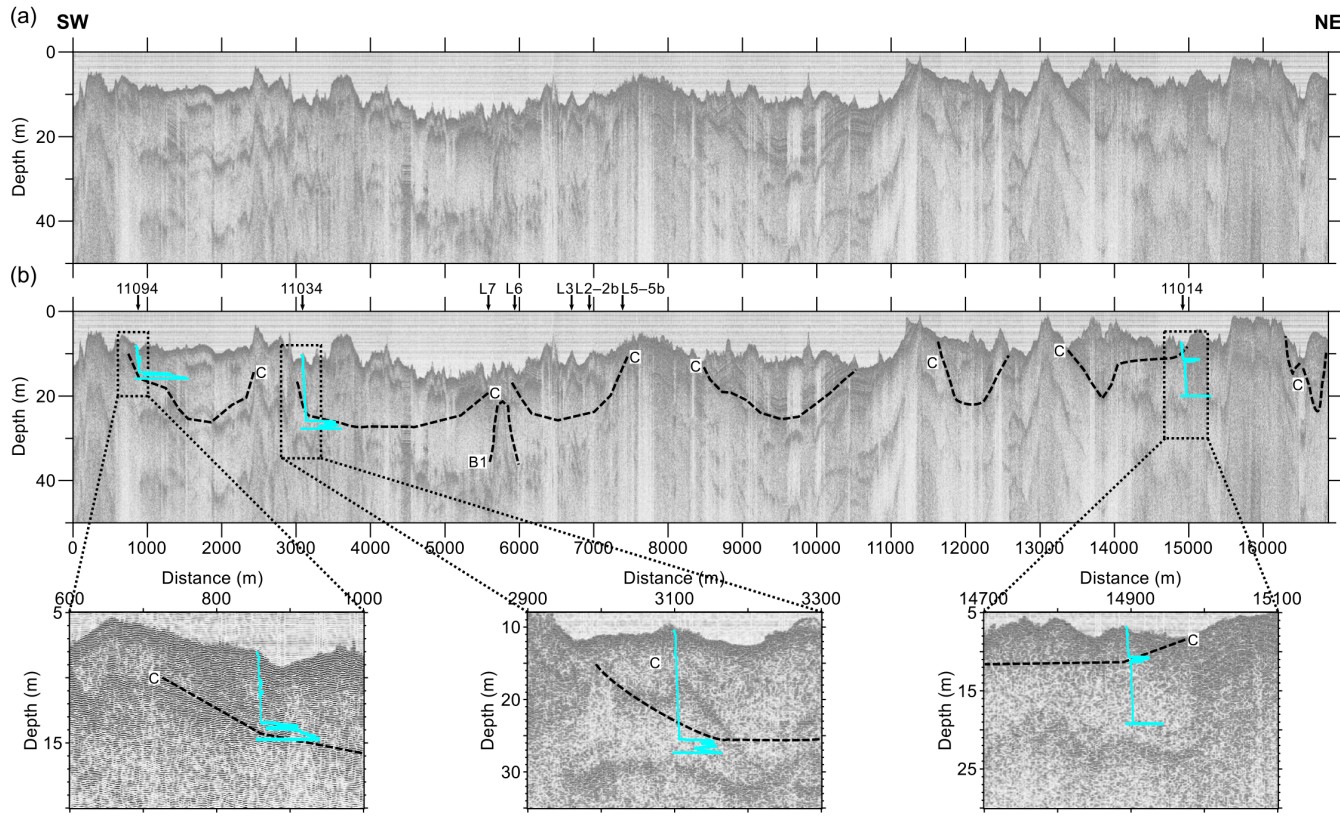

**Figure 9.** Processing results of the single-channel (3.5 kHz echo sounder) river seismic data (ⓒ SGU). **(a)** Seismic processed section. **(b)** Interpreted seismic section including total sounding data (BGA, 2018) from boreholes 11014 (ranging from 0 to 7.2 kN, distance to the seismic line 34 m), 11034 (ranging from 0 to 9.2 kN, distance to the seismic line 20.5 m), and 11094 (ranging from 0 to 11.2 kN, distance to the seismic line 36 m), all in blue. Close-ups for each borehole are shown at the bottom part of the figure. C: filled channels and B1: bedrock represented by dashed black lines. The positions of the land seismic lines that intersect this line are indicated on top of the section. Scale 1H:57V.





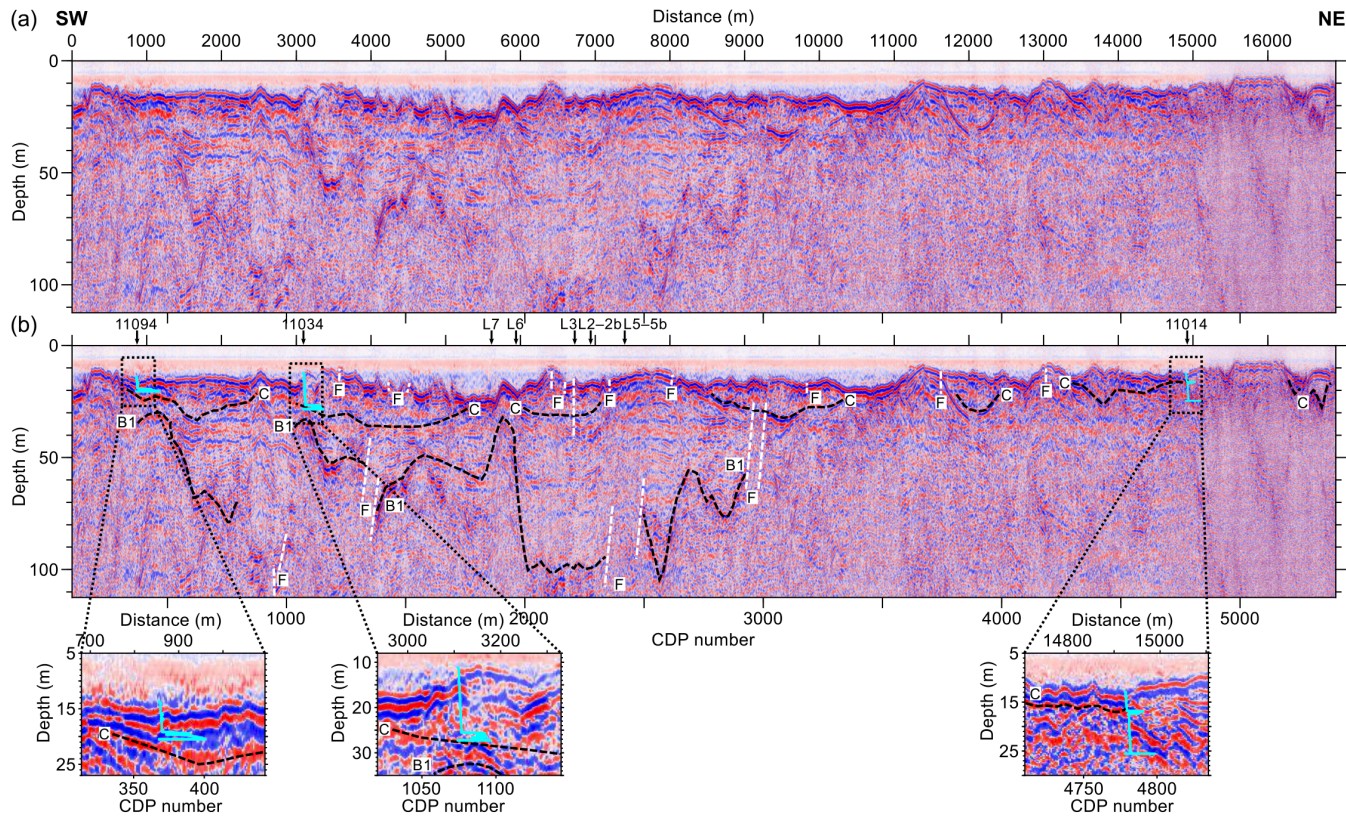

**Figure 10.** Processing results of the six-channel river seismic data (© SGU). **(a)** Seismic processed section. **(b)** Interpreted seismic section including total sounding data (BGA, 2018) from boreholes 11014 (ranging from 0 to 7.2 kN, distance to the seismic line 45 m), 11034 (ranging from 0 to 9.2 kN, distance to the seismic line 10 m), and 11094 (ranging from 0 to 11.2 kN, distance to the seismic line 25.7 m), all in blue. Close-ups for each borehole are shown at the bottom part of the figure. C: filled channels and B1: bedrock represented by dashed black lines, and F: fractured or disturbed materials (no reflection continuity) represented by a dashed white line. The positions of the land seismic lines that intersect this line are indicated on top of the section. Scale 1H:30V.



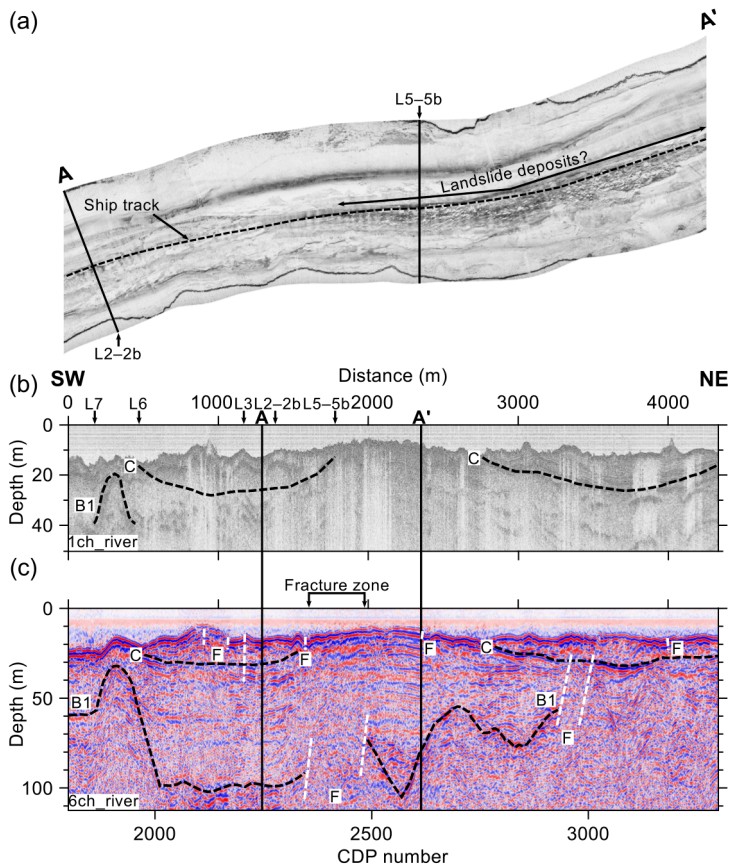

**Figure 11.** A section of the river seismic lines and side-scan sonar data (© SGU). **(a)** Side-scan sonar data corresponding to the profile AA' shown on top of the section in Figure 11b. The track followed by the ship, the position of the possible landslide deposits and the land seismic lines that intersect this area are indicated. **(b)** Interpreted single-channel seismic section. Scale 1H:12V. **(c)** Interpreted six-channel seismic section. Scale 1H:12V. C: filled channels, B1: bedrock, F: fractured or disturbed materials (no reflection continuity).





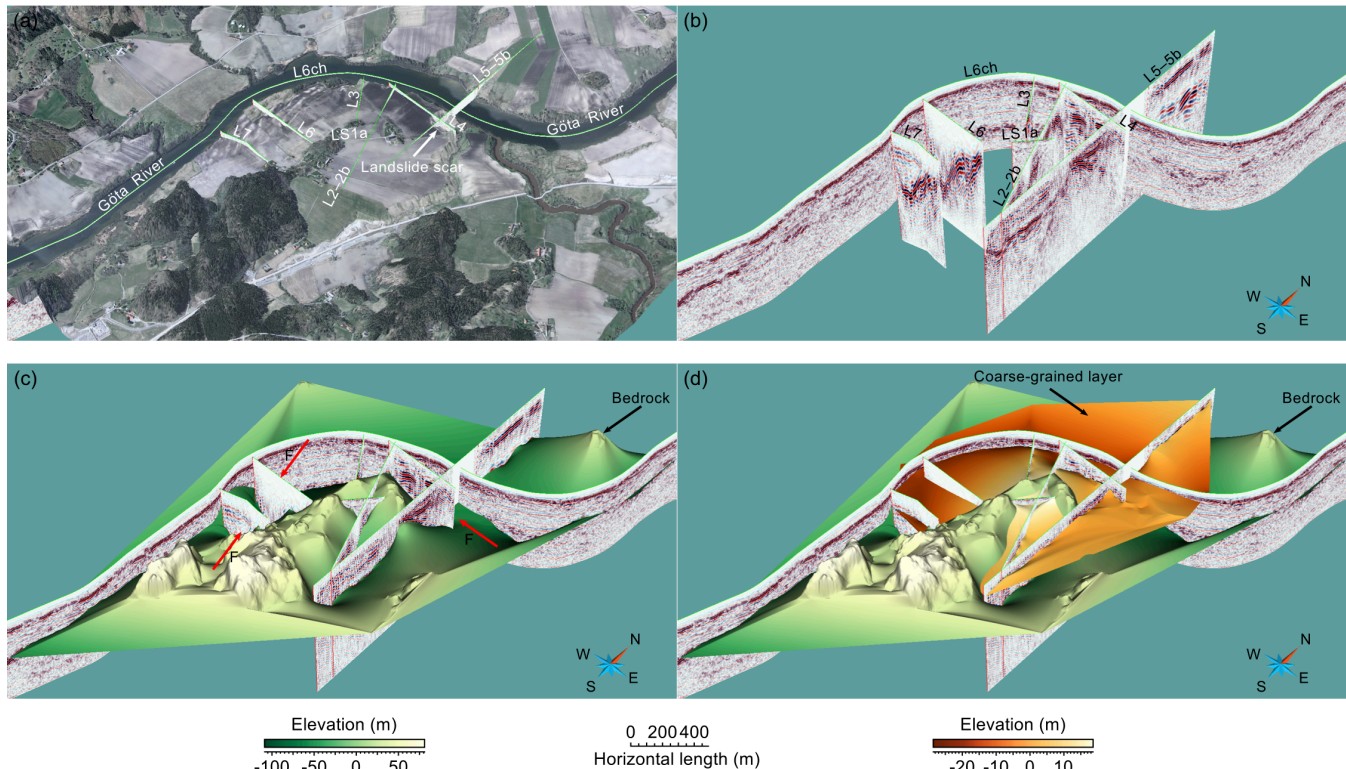

**Figure 12.** 3D Modelling of the subsurface at the survey site (south view). **(a)** Aerial photo projected on the LiDAR elevation surface (© Lantmäteriet) of the study area with the positions of the land and river seismic lines along the Göta River. **(b)** 3D view of the land and river seismic processed lines. **(c)** Elevation of the bedrock surface (F indicates the position of interpreted faults, also observed in Fig. 2a). **(d)** Elevation of the coarse-grained layer and bedrock surfaces. The elevation values for both surfaces and the horizontal scale are shown at the bottom of the figure.



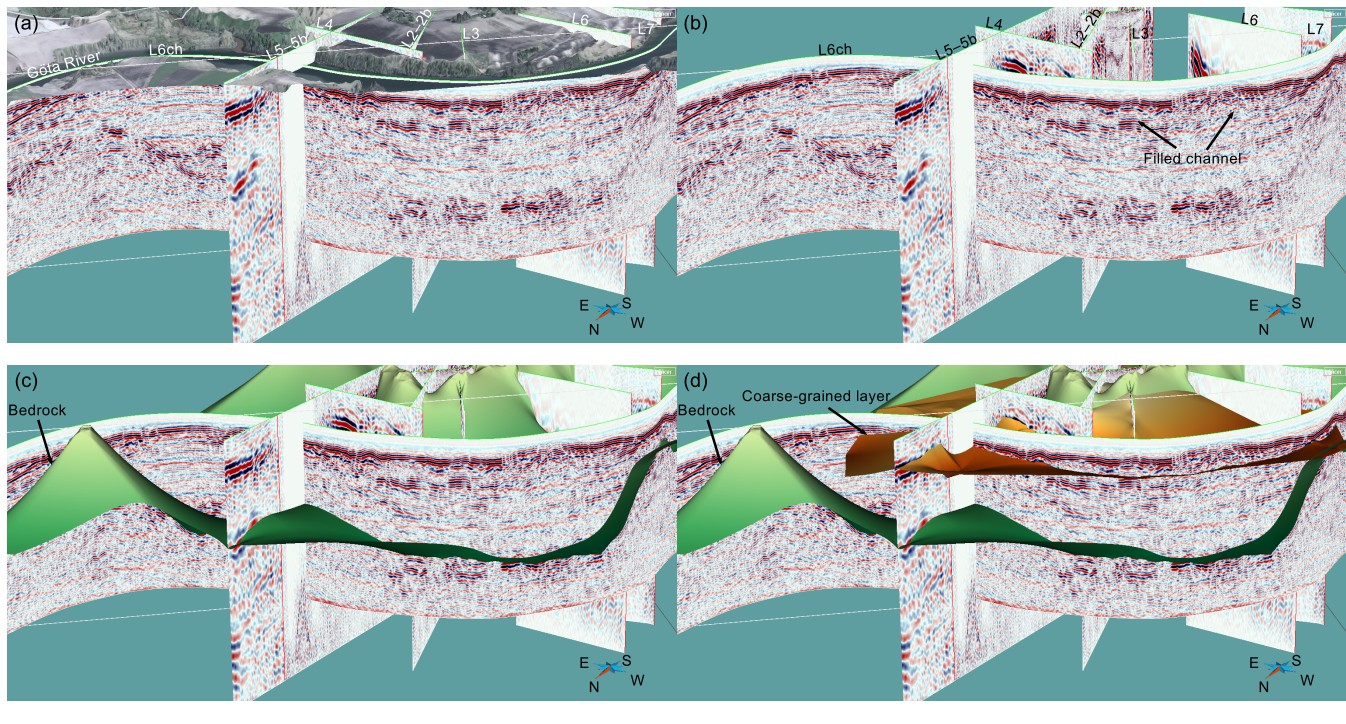

0  150 300
Horizontal length (m)

**Figure 13.** 3D views of the seismic data and key surfaces in the study area. **(a)** Aerial photo projected on the LiDAR elevation surface (© Lantmäteriet) with the positions of the land and river seismic lines along the Göta River. **(b)** 3D view of the land and river seismic processed lines. **(c)** Elevation of the bedrock surface. **(d)** Elevation of the coarse-grained layer and bedrock surfaces. Note the coincidence of the coarse-grained layer surface with the delineation of one of the filled channels (see Figures 9, 10 and 11). Observe also the undulated bedrock surface between lines 6 and 7 located on the right side of the figure. See elevation values for both surfaces in Fig. 12. The horizontal scale is shown at the bottom part of the figure.





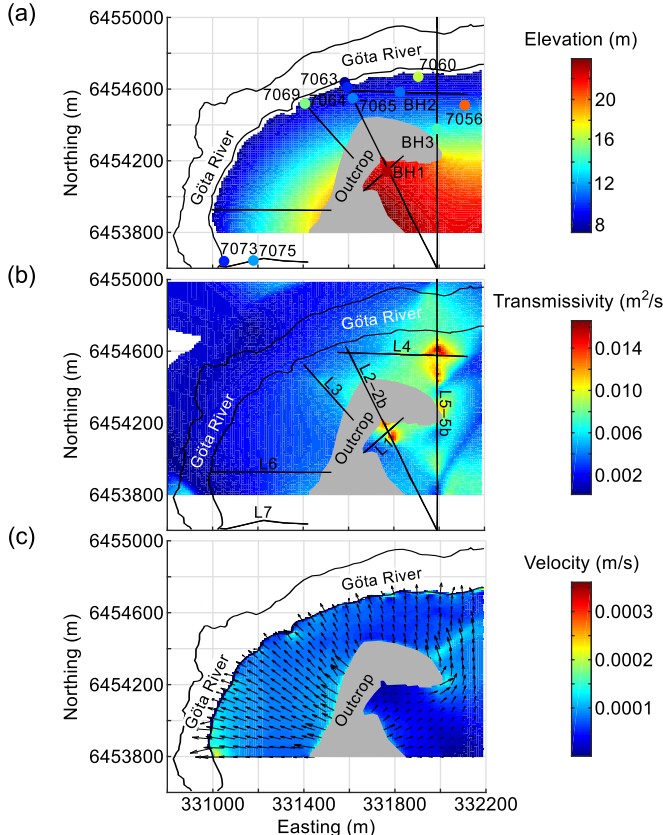

**Figure 14.** Hydrological modelling of a selected region in the study area. **(a)** Water table elevation. **(b)** Transmissivity distribution. **(c)** Mean groundwater velocity distribution and vector field of the Darcy flow. Range of values on the right side of each figure. The position of the Göta River, outcrop area, land seismic lines (Malehmir et al., 2013a, 2013b; Salas-Romero et al., 2015) and boreholes (BGA, 2018; Salas-Romero et al., 2015) used in the modelling are indicated in the figure.



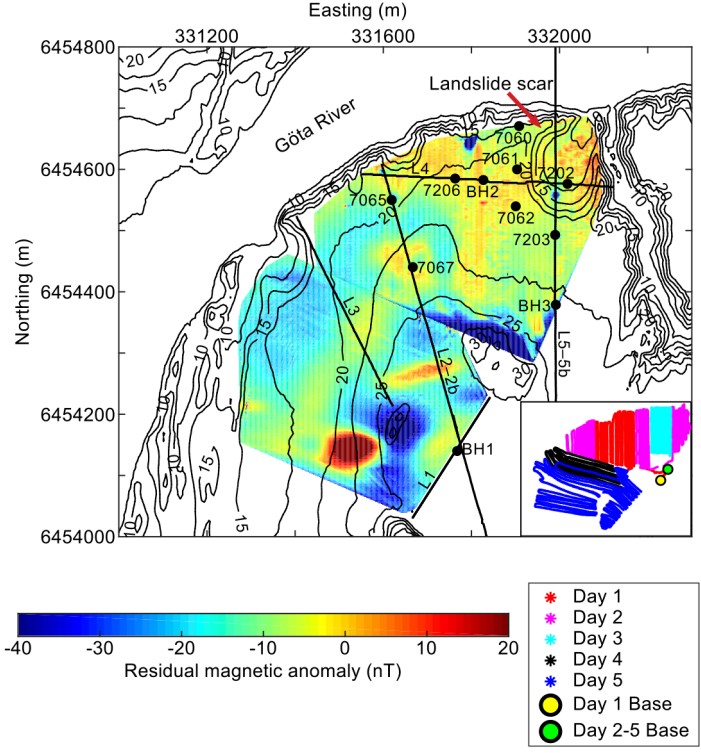

**Figure 15.** Residual magnetic anomaly data plotted on the contour map of the LiDAR data (© Lantmäteriet). The position of the land seismic lines (Malehmir et al., 2013a, 2013b; Salas-Romero et al., 2015) and boreholes (BGA, 2018; Löfroth et al., 2011; Salas-Romero et al., 2015) used for analysing the magnetic data are shown in the map. In the bottom right, a sketch showing the measurement and base station positions for each day can be found.

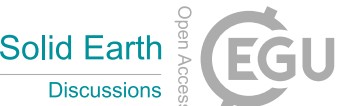







**Figure 16.** Side-scan sonar data (© SGU) overlying the bathymetric data (© SGI) on selected segments of the Göta River. See position of the figures in Fig. 2c. **(a)** Cross section next to the study area, showing also the position of line 5–5b. Scale 1H:8V. The area delineated with a dashed red line shows accumulated material at the bottom of the river. **(b)** Elevation along profile AA'. **(c)** Slope (absolute values) along profile AA'. **(d)** Elevation along profile BB'. **(e)** Slope (absolute values) along profile BB'. **(f)** Cross section of the river. Scale 1H:3V. The area delineated with a dashed red line shows accumulated material in the eastern margin. The area delineated with a dashed blue line shows a probable subaquatic landslide scar. **(g)** Elevation along profile AA'. **(h)** Slope (absolute values) along profile AA'. **(i)** Elevation along profile BB'. **(j)** Slope (absolute values) along profile BB'. **(k)** Cross section of the river. Scale 1H:4V. The area delineated with a dashed red line shows accumulated material at the bottom of the river. **(l)** Elevation along profile AA'. **(m)** Slope (absolute values) along profile AA'.