# Peer review of "Subsurface structures of a quick-clay sliding prone area revealed using land-river reflection seismic data and hydrogeological modelling"

_Solid Earth, 2019_

## Referee Comment (RC1) · Anonymous Referee #1 · 27 Feb 2019

SUMMARY

The paper attempts the challenging task of integrating a large collection of geological and geophysical data to derive new insights on landslide formation in quick clays for a specific study area. Primarily using seismic reflection data, the study results in the mapping of the bedrock surface, and an overburden aquifer that may be associated with quick clay formation and landslide activity. However, the paper is poorly organized to the point that the results and their importance are obfuscated.

In particular, the authors do not clearly distinguish between established data and interpretation, and the links between the observed data and the process-based conclusions are not effectively demonstrated. The paper would be more impactful if, for example, the authors succinctly identified previously-published data and interpretations, and then demonstrated how new data and insights build on past knowledge to support their objectives. In this fashion, more focus could be placed on the novel aspects of the paper including the hydrogeological modelling, and testing the hypotheses (or supporting the interpretations) of aquifer-driven leaching and sliding mechanism.

Some editing for English is required throughout the paper.

DETAILS

p2,L2: Revise sentence.

p2,L4: Reference required.

p2: Consider revising/restructuring the introduction to improve focus and logical flow of ideas. Focus switches from geophysics to hydrogeology to site specific conditions all in the same paragraph without any transitions or linking of ideas.

p3: Consider reorganization of material into separate "Intro" and a "Study Area" sections. Geological maps, shot locations, and LiDAR data are not really introductory material. Clearly differentiate between legacy data, and new data. It is not apparent what the new contributions are.

p3,L13: The slide is of interest because it is in the middle of the study area?!

p.3,L27: Is this the hypothesis being presented? The objective of the paper is not yet clear at this point.

p.4: If all of these data sets are legacy data sets, they should be well described in the cited references. Much of this information does not seem necessary to support the objectives.

p.7: Velocity analysis was the "most important step" but receives less discussion than routine operations. How does the time-depth conversion velocity compare to the results of the velocity analysis?

p.7,L32: What is the nature of S1? In particular, does it have finite thickness relative to the wavelength? You refer to it as a layer, an interface and a horizon. How are you picking it? Because it could be interpreted as a compacted top of a reflection package, which seems to be supported by the mag. suc. that increases below S1 and stays high. More discussion of the interpretation logic and reflection facies is necessary. Core?

p.8,L19: If there is core, how come lithological logs are not part of the analysis?

p.9,L32: It looks like a decrease in penetration resistance.

p11,L6: The landslide scar is not apparent in Fig.11.

p.11,L11: The paper seems to have been comparing previous studies the whole time.

p.11,L15: It is unclear how this proposed mechanism works with infiltration "through outcrops and fracture zones" and how this is related to the coarse-grained layer. Consider more development of the hydrogeological conceptual model, and then provide supporting evidence.

p.11,L25: This is well below your quoted resolution. You have not show any frequency spectra, but it is likely well below quarter-wavelength as well.

p.12,L13: Where are the total sounding data with top and bottom of S1? The nature of S1 and the coarse layer (deposition, thickness, etc.) has not been discussed up to this point.

p.12,L13: Where are the data that go into this most important interpolation? How are you handling the multiple interpreted faults?

p.13,L13: There are more boreholes in Fig.2b than in Fig.14. Need to distinguish water wells from other holes.

p.13,L14: It is hard to tell from the figure, but it appears that 3 of 7 boreholes in the model domain (excluding the southern holes) are not fit by the model. This is a lot, and is attributed to "only one measurement" but these holes have not been distinguished from any of the other holes (do they have two measurements?) and the possibility of the model simply being wrong needs to be addressed. Consider presenting and discussing water well data.

p.13,L32: This needs to be explained.

p.14,L5: Explain this. You have "calibrated" the recharge area and the recharge transmissivity which will have a direct trade-off with the required recharge. The modelling requires some sensitivity analysis.

p.14,L8: What data?

p.14.L17: Groundwater velocity of 0.00015m/s or ~13m/day is very fast.

p.14: Consider showing and analyzing the mag data with the rest of the data in the data section as opposed to in the discussion section.

p15,L21: Should be easy enough to test with a multivariate regression of mag, T and depth - or some combined variable of T and depth.

p.17: In my opinion, many of the conclusions from L16 down (and in the abstract) remain conjecture. The catchment area is prescribed without validation, it is not clearly demonstrated that there is aquifer-driven leaching, or that the coarse layer is a sliding surface, and the nature of the mag. anomaly is an interpretation.

Fig.1: Demonstrate value of this figure.

Fig.2: Shorten caption. Do not repeat what is stated in the text or what is evident from the figure (such as the legend). Improve figure clarity. Much of the text and symbols on the map are not easily legible.

Figs.3&4: Remove repetitive material from captions.

Figs.5&6: The captions are far too long and complicated. The borehole logs require scales and labels (other figures also).

Fig.12: While visually impressive, perspective images are not good for evaluating interpolated surfaces, particular with respect to potential bias introduced by spatially non-uniform data such as sparse boreholes combined with dense points along seismic lines. Contour plots should be used for analysis. What about coarse layer thickness?

Fig.13: b and c add nothing.

Table 2: For all but the LiDAR perhaps, these values are nominal resolutions, and the spatial sampling interval is not indicative of the resolution.

---

## Referee Comment (RC2) · Adam Booth (Referee) · 25 Mar 2019

Summary

This is an ambitious paper that shows the strengths of combining multiple data sources together. As the authors point out in the introduction, the analysis is highly multi-disciplinary and multi-methodological, and the hydrological modelling draws on a diverse set of data. Having less expertise in hydrological modelling, most of my comments pertain to the treatment of the geophysical data and the general format of the

paper.

I have two main criticisms of the paper:

a) At times, it seems a little lengthy. An aspect of this is the length of some of the paragraphs (Section 4.1, for example, is a monster which spans Pages 8 and 9!); break these up a bit to improve the appreciation of your process.

b) Sometimes the interpretation of the seismic data is also over-long, but also over-interpreted. I list some specific examples below (Points 13-16), but the key point is that not all of the seismic observations appear to have significance in the model – so I think you should restrict the discussion of the interpretation to the most relevant parameters. A full interpretation could go into supplementary material, although (see below) I'd suggest that some of this is over-interpreted anyway.

With such streamlining, the objectives of your paper will be more understandable and its significance therefore more appreciable.

Specific points

1. Title. This indicates that you reveal subsurface structures with modelling, but I'm not sure this is what you mean. Presumably, the structures you image in the geophysical data help constrain the model? A title like "Hydrological modelling of a quick-clay vulnerable area, constrained with geophysical data" would be more informative?

2. Abstract. For all the numerical analysis in your paper, the abstract contains no numbers. Can you add some in? e.g., some highlights from the geophysical dataset, and some of the hydrological parameters you use and model?

3. P1L24: "sensitive" – to what? Makes it sound a bit like they are emotional!

4. P2L4: Explain the terminology "sensitivity higher than 50"... Is there a unit or a reference system here?

5. P2L7: Surely there's no need to separate "geotechnics, geophysics or geology" out?

Aren't they're all "geoscience"?

6. Section 2.1 (and throughout): You variously refer to your seismic lines by number, or by the source acquisition method. I found this very confusing, trying to remember what method was used on what line, and would prefer that you stick to the numerical reference throughout. The table usefully informs what source was used anyway.

7. P5L25: No need to say "reflected sound waves": if they are transmitted as sound waves, they'll come back as sound waves!

8. P6L16: What velocity from the 800-4000 m/s range did you settle on? It doesn't seem to be listed anywhere.

9. P7L3: Why the different mute definition for the wireless data?

10. P7L23: What is this absolute value of error with respect to? Give it as a fraction of the typical target depth?

11. P7L25: To help with the interpretation, it might be worth tabulating the expected response of the different geologies you interpret in each geophysical dataset. Even just listing the range of seismic velocities and resistivities you might expect would help your data description.

12. P8L5: Are you implying that the borehole is 0.02 m, or 0.02 *km* away from the seismic line? If it really is 0.02 m, then it hardly seems worth reporting this, and you could just say that the borehole lies on the seismic line.

13. P8L7: The interpreted faults are not really clear, and it seems an over-interpretation particularly since refraction static corrections were not applied. Could near-surface anomalies be the origin of the discontinuities and misalignments that you claim? In any case are the faults and damage zones critical to your model? It seems to me that you could be much more tentative in interpreting them, without damaging any parameter in your model.

14. P8L8: You don't get a lot of reliable ray coverage in the refraction tomography to really talk about the velocities below reflection B1. I agree that your velocities above this horizon are likely reliable, and you do point out that they have velocities consistent with coarse-grained, saturated sediment. However, in general, I find the resistivity data (Figure 5d, Figure 6e) to provide the much more compelling evidence of a bedrock underburden.

15. P9L2: On what grounds to you interpret a kinematic response from the seismic data? You see dipping horizons, but I don't see how you can say that this represents s slip surface.

16. P9L6: I would suggest that it is beyond the capability of travel-time inversion to resolve boulders, as you claim here. I might expect that they could appear as diffractions in the seismic section, or high-resistivity anomalies, but I don't believe that the tomography would be sensitive to them. Furthermore, this over-interpretation doesn't actually appear to influence any parameterisation of your model, so the paper wouldn't be damaged if you said that your tomography has some unexplained velocity artefacts.

17. P11L20: You suggest that the seismic data shows a higher-resolution delineation of the bedrock/sediment contact, but you wouldn't be able to make this interpretation if it wasn't for the sum total of your datasets! It therefore seems unnecessary to make this assertion when you draw on inferences from all of your data – it doesn't matter which is best! Indeed, this whole section could be considered for removal as it's not clear to me that you are presenting a different hypothesis to one that has been previously postulated. It will always be the case that the use of multiple data sources leads to an improved interpretation.

Figures

1. In the interpretation of Figure 9, you correctly point out in the main text that you are prone to mistaking multiples for genuine reflections. You appear to avoid multiples, except (potentially) for the interpretation between ∼800-2500 m in Figure 9b. Can you

be sure that this hasn't been misinterpreted? Also, the inset figures here add very little: the data look very fuzzy, so much so that the logs don't appear to correlate with anything at all.

2. There are potentially too many figures in the paper, and 12 and 13 could be earmarked for removal as they're not very clear partly because of the limited quality of the seismic data. Could they be moved into supplementary material instead? Equally, once the interpretation is streamlined, I don't think that all the seismic lines need to be included.

3. Some figure captions need to be reduced in length, typically those relating to the seismic lines (Figures 5,6,8).

---

## Author Comment (AC1) · 5 May 2019

We thank the anonymous reviewer for the critical and useful comments. We have addressed all the specific comments in our revised manuscript, as detailed below.

(Anonymous Referee #1) The paper attempts the challenging task of integrating a large collection of geological and geophysical data to derive new insights on landslide formation in quick clays for a specific study area. Primarily using seismic reflection data, the study results in the mapping of the bedrock surface, and an overburden aquifer

that may be associated with quick clay formation and landslide activity. However, the paper is poorly organized to the point that the results and their importance are obfuscated. In particular, the authors do not clearly distinguish between established data and interpretation, and the links between the observed data and the process-based conclusions are not effectively demonstrated. The paper would be more impactful if, for example, the authors succinctly identified previously-published data and interpretations, and then demonstrated how new data and insights build on past knowledge to support their objectives. In this fashion, more focus could be placed on the novel aspects of the paper including the hydrogeological modelling, and testing the hypotheses (or supporting the interpretations) of aquifer-driven leaching and sliding mechanism.

(Authors) We thank the reviewer for the comments, which have been useful for improving our manuscript. We have followed the advice and put more emphasis on distinguishing the previous data and our new/current contributions, and relate the last ones to the conclusions.

(Anonymous Referee #1)

Some editing for English is required throughout the paper.

(Authors) A native speaker has gone through the revised version and we hope this problem is fixed. We have taken additional steps to improve the readability and flow of the text.

(Anonymous Referee #1) p2, L2: Revise sentence.

(Authors) The sentence has been modified.

The new sentence is: 'The presence of quick clays can only be confirmed using geotechnical site and laboratory investigations enabling estimation of the sensitivity (Rankka et al., 2004). The sensitivity is defined as the ratio of undrained undisturbed to remoulded shear strength.'

(Anonymous Referee #1) p2, L4: Reference required.

(Authors) A new reference has been added now. • Karlsson, R. and Hansbo, S. (1989). Soil classification and identification. Byggforskningsrådet Document D8:1989. Stockholm.

(Anonymous Referee #1) p2: Consider revising/restructuring the introduction to improve focus and logical flow of ideas. Focus switches from geophysics to hydrogeology to site specific conditions all in the same paragraph without any transitions or linking of ideas.

(Authors) We have followed this advice and have restructured the Introduction and ordered the objectives around this, creating a new subsection that clarifies the introductory materials.

(Anonymous Referee #1) p3: Consider reorganization of material into separate "Intro" and a "Study Area" sections. Geological maps, shot locations, and LiDAR data are not really introductory material. Clearly differentiate between legacy data, and new data. It is not apparent what the new contributions are.

(Authors) As mentioned, we have modified the Introduction and created a new section called Study Area. This new section includes the data that do not belong to the Introduction, such as the geological and topographical data. We also have reworked the text to better describe what legacy data are and what new contributions are.

(Anonymous Referee #1) p3, L13: The slide is of interest because it is in the middle of the study area?!

(Authors) We have modified the sentence that makes reference to the landslide scar in the study area.

(Anonymous Referee #1) p.3, L27: Is this the hypothesis being presented? The objective of the paper is not yet clear at this point.

(Authors) No, this is not the hypothesis being presented in this paper. A reference has been added to clarify this. The objectives were listed on p. 3, starting in line 32.

(Anonymous Referee #1) p.4: If all of these data sets are legacy data sets, they should be well described in the cited references. Much of this information does not seem necessary to support the objectives.

(Authors) All these data are being used within this study for achieving the objectives. We think it is reasonable to reproduce this information so that the article is as complete and as "stand alone" as possible, avoiding too much cross-referencing.

(Anonymous Referee #1) p.7: Velocity analysis was the "most important step" but receives less discussion than routine operations. How does the time-depth conversion velocity compare to the results of the velocity analysis?

(Authors) Velocity analysis was an important step in the cabled part of line 5–5b, but not the only processing step that improved the quality of the final seismic section (the text has been modified accordingly). The time-depth conversion velocity (1500 m/s) is within the range of velocities picked during the velocity analysis (1100-1600 m/s) and is also consistent with fully saturated clay, which has velocity similar to water. Velocity analysis had to take care of the dip component (dip-velocity dependent stacking velocity) but this is not required for time-to-depth conversion.

(Anonymous Referee #1) p.7, L32: What is the nature of S1? In particular, does it have finite thickness relative to the wavelength? You refer to it as a layer, an interface and a horizon. How are you picking it? Because it could be interpreted as a compacted top of a reflection package, which seems to be supported by the mag. suc. that increases below S1 and stays high. More discussion of the interpretation logic and reflection facies is necessary. Core?

(Authors) The text has been modified to clarify that S1 represents the top of the coarse-grained layer, i.e. an interface. Our earlier publication in the journal of Landslides and a previous study by Malehmir et al. (2013) already identified S1 and discuss the nature of this layer, using core samples and other evidence.

[Figure]

(Anonymous Referee #1) p.8, L19: If there is core, how come lithological logs are not part of the analysis?

(Authors) In the previous paper published in Landslides by Salas-Romero et al. (2015) ('Identifying landslide preconditions in Swedish quick clays–insights from integration of surface geophysical, core sample- and downhole property measurements'), grain size distributions are shown for the core samples of the three boreholes (also soil textures are estimated with the natural gamma radiation data). We believe that these results are good enough for describing the core material as they are soils and their appearance is difficult to quantify in a visual inspection (the cores collected in the three boreholes drilled in the study area were visually inspected when samples were collected for the different laboratory measurements). We use some of the core information from our previous work for supporting the new data, but we prefer not to repeat too much in this manuscript.

(Anonymous Referee #1) p.9, L32: It looks like a decrease in penetration resistance.

(Authors) We have removed borehole 7073 as the increase in penetration resistance is unclear, but we keep borehole 7075 because it does show an increase in penetration resistance at the interface S1. It may not be visible at the figure's size but the reference can be checked online.

(Anonymous Referee #1) p11, L6: The landslide scar is not apparent in Fig.11.

(Authors) We did not mean that the landslide scar is visible in Fig. 11, but that the landslide scar is located in that position and that the deposits located in the river may be related to it. We have rephrased the sentence and made it clearer to the reader.

(Anonymous Referee #1) p.11, L11: The paper seems to have been comparing previous studies the whole time.

(Authors) We have reworked this subsection to include only new contributions when comparing with previous studies.

(Anonymous Referee #1) p.11, L15: It is unclear how this proposed mechanism works with infiltration "through outcrops and fracture zones" and how this is related to the coarse-grained layer. Consider more development of the hydrogeological conceptual model, and then provide supporting evidence.

(Authors) The hydrological model has been improved and we hope that it now provides sufficient information about the infiltration mechanism.

(Anonymous Referee #1) p.11, L25: This is well below your quoted resolution. You have not show any frequency spectra, but it is likely well below quarter-wavelength as well.

(Authors) We are aware that our data resolution cannot distinguish such a thin layer. Our intention was to mention that we have checked other borehole data to the north and south of the study area and that the coarse-grained layer can be found in many of them but in some places can be thinner than in our study area.

(Anonymous Referee #1) p.12, L13: Where are the total sounding data with top and bottom of S1? The nature of S1 and the coarse layer (deposition, thickness, etc.) has not been discussed up to this point.

(Authors) The total sounding data are not included in this manuscript as a lot of information is already present in it. Nevertheless, we provide references about where to find this material, available for any person providing the area information and borehole id. We discussed largely about the nature of the coarse-grained layer in our paper published in Landslides (reference included in the text). Thus, we do not think it is necessary to discuss it again.

(Anonymous Referee #1) p.12, L13: Where are the data that go into this most important interpolation? How are you handling the multiple interpreted faults?

(Authors) The boreholes used in the interpolation are now included in a new figure that shows the surface contours. The LiDAR data are shown partially in Fig. 2b. The

interpreted faults have not been used for the interpolation of the elevation surfaces.

(Anonymous Referee #1) p.13, L13: There are more boreholes in Fig.2b than in Fig.14. Need to distinguish water wells from other holes.

(Authors) We have improved the maps and clarified this information distinguishing between the wells used in the hydrological modelling and the others in Fig. 2 and in the new figure showing the surface contours, as mentioned before.

(Anonymous Referee #1) p.13, L14: It is hard to tell from the figure, but it appears that 3 of 7 boreholes in the model domain (excluding the southern holes) are not fit by the model. This is a lot, and is attributed to "only one measurement" but these holes have not been distinguished from any of the other holes (do they have two measurements?) and the possibility of the model simply being wrong needs to be addressed. Consider presenting and discussing water well data.

(Authors) We provide a table with information about the pore pressure and well data and sort them by depths near the estimated coarse-grained layer (lower aquifer) versus shallower measurements in the fissured near surface clay (upper aquifer). With this distinction in place our simple model manages to approach the former type of values with an RMSE of 0.5 m.

(Anonymous Referee #1) p.13, L32: This needs to be explained.

(Authors) We have intensively reworked and extended that paragraph to make it clearer to the reader.

(Anonymous Referee #1) p.14, L5: Explain this. You have "calibrated" the recharge area and the recharge transmissivity which will have a direct trade-off with the required recharge. The modelling requires some sensitivity analysis.

(Authors) A new automated calibration procedure gave better estimates of the optimal values and a better understanding of the model near that minimum (an updated figure was prepared). As can be understood, the two parameters can partially compensate

for each other, which leaves some uncertainty in their estimate. Hydraulic tests and/or tracer tests would be required to improve the estimates of the model parameters or to elaborate on the model structure. This single layer hydrogeological model was built almost as a back-of-the-envelope scoping calculation and will not bring positive proof of the validity of the leaching assumption as the cause of quick-clay formation. The intention was to provide a quick plausibility check for this area.

(Anonymous Referee #1) p.14, L8: What data?

(Authors) We rephrased this sentence to make it clearer. The statement is about the pore pressure data.

(Anonymous Referee #1) p.14, L17: Groundwater velocity of 0.00015m/s or âĹij13m/day is very fast.

(Authors) We agree with the reviewer. This velocity obviously shows some limitations of the simple hydrogeological model that isolate the coarse-grained layer.

(Anonymous Referee #1) p.14: Consider showing and analyzing the mag data with the rest of the data in the data section as opposed to in the discussion section.

(Authors) We have considered it and we still believe that the magnetic data are better discussed in the Discussion, subsection about the morphology of the Göta River valley. The data in the Results section are subsurface data describing the subsurface structures, whereas the magnetic data are superficial data.

(Anonymous Referee #1) p15, L21: Should be easy enough to test with a multivariate regression of mag, T and depth - or some combined variable of T and depth.

(Authors) We thank the reviewer for this useful advice. This test helped interpreting the results.

(Anonymous Referee #1) p.17: In my opinion, many of the conclusions from L16 down (and in the abstract) remain conjecture. The catchment area is prescribed without val-
idation, it is not clearly demonstrated that there is aquifer-driven leaching, or that the coarse layer is a sliding surface, and the nature of the mag. anomaly is an interpretation.

(Authors) The conclusions have been modified in order to add the new information obtained after this revision.

(Anonymous Referee #1) Fig.1: Demonstrate value of this figure.

(Authors) We believe that it is interesting to show the potential for destruction of this type of landslides near the same river in an area very close to the south of our study area. The current risk of landslides is medium-high in the study area, so we think that the picture is illustrative of a possible scenario.

(Anonymous Referee #1) Fig.2: Shorten caption. Do not repeat what is stated in the text or what is evident from the figure (such as the legend). Improve figure clarity. Much of the text and symbols on the map are not easily legible.

(Authors) We followed this advice, and shortened the caption and improved the figure clarity.

(Anonymous Referee #1) Figs.3&4: Remove repetitive material from captions.

(Authors) The captions have been shortened.

(Anonymous Referee #1) Figs.5&6: The captions are far too long and complicated. The borehole logs require scales and labels (other figures also).

(Authors) The captions have been shortened. We do not think that we require scales or labels for the borehole logs. Precisely, we included close-ups for each borehole in every figure for helping to visualize them. The maximum and minimum values for every log are included in the caption of the figure. We believe that adding scales to these figures will worsen their clarity. The same information and more details can be found in our previous work published in Landslides and in the SGI website.

(Anonymous Referee #1) Fig.12: While visually impressive, perspective images are not good for evaluating interpolated surfaces, particular with respect to potential bias introduced by spatially non-uniform data such as sparse boreholes combined with dense points along seismic lines. Contour plots should be used for analysis. What about coarse layer thickness?

(Authors) We followed this advice and added a new figure that shows the contour plots for the elevation surfaces. The perspective images are now included in the supplementary material for not increasing the manuscript length. We prefer to keep the perspective images because they give a good perspective of the relationship between the surfaces, the river, and possible fault zones. The coarse-grained layer thickness is an estimation that may include more uncertainties than the delineation of the top of the layer. Figure 14b (transmissivity) shows essentially the layer thickness as T=K*thickness, K=const.

(Anonymous Referee #1) Fig.13: b and c add nothing.

(Authors) These figures have been removed.

(Anonymous Referee #1) Table 2: For all but the LiDAR perhaps, these values are nominal resolutions, and the spatial sampling interval is not indicative of the resolution.

(Authors) We agree with the reviewer and added more information to the table for clarifying these terms.

---

## Author Comment (AC2) · 5 May 2019

We thank A. Booth for the critical and useful comments. We have addressed all the specific comments in our revised manuscript, as detailed below.

(A. Booth Referee #2) This is an ambitious paper that shows the strengths of combining multiple data sources together. As the authors point out in the introduction, the analysis is highly multi- disciplinary and multi-methodological, and the hydrological modelling draws on a di- verse set of data. Having less expertise in hydrological modelling, most

[Figure]

of my comments pertain to the treatment of the geophysical data and the general format of the paper.

(Authors) We thank the reviewer for his comments, which have been very useful for improving our manuscript. We hope the general format of the manuscript (length and structure) is improved after this revision.

(A. Booth Referee #2) a) At times, it seems a little lengthy. An aspect of this is the length of some of the paragraphs (Section 4.1, for example, is a monster which spans Pages 8 and 9!); break these up a bit to improve the appreciation of your process.

(Authors) We agree with the reviewer and have reworked different sections and moved figures to the supplementary material for improving the readability.

(A. Booth Referee #2) b) Sometimes the interpretation of the seismic data is also over-long, but also over- interpreted. I list some specific examples below (Points 13-16), but the key point is that not all of the seismic observations appear to have significance in the model – so I think you should restrict the discussion of the interpretation to the most relevant parameters. A full interpretation could go into supplementary material, although (see below) I'd suggest that some of this is over-interpreted anyway.

(Authors) We followed this advice and reduced the interpretation section. Some of the figures are now included in the supplementary material.

(A. Booth Referee #2) With such streamlining, the objectives of your paper will be more understandable and its significance therefore more appreciable.

(Authors) We thank for the comments and hope that the objectives of the manuscript are now clearer.

(A. Booth Referee #2) Title. This indicates that you reveal subsurface structures with modelling, but I'm not sure this is what you mean. Presumably, the structures you image in the geophysical data help constrain the model? A title like "Hydrological modelling of a quick-clay vulnerable area, constrained with geophysical data" would be

more informative?

(Authors) We thank the reviewer for the suggestion. We agree to change the title to 'Hydrogeological modelling of a quick-clay vulnerable area, constrained with geophysical data'.

(A. Booth Referee #2) Abstract. For all the numerical analysis in your paper, the abstract contains no numbers. Can you add some in? e.g., some highlights from the geophysical dataset, and some of the hydrological parameters you use and model?

(Authors) We have added some numerical information to the abstract.

(A. Booth Referee #2) P1L24: "sensitive" – to what? Makes it sound a bit like they are emotional!

(Authors) This type of clays is usually described using this geotechnical terminology.

(A. Booth Referee #2) P2L4: Explain the terminology "sensitivity higher than 50"... Is there a unit or a reference system here?

(Authors) The definition of sensitivity is given in this page in line 3. It is the ratio of the undrained undisturbed shear strength to the remoulded shear strength, thus there is no unit.

(A. Booth Referee #2) P2L7: Surely there's no need to separate "geotechnics, geophysics or geology" out? Aren't they're all "geoscience"?

(Authors) We agree with the reviewer and have modified the sentence for using only the geoscience term.

(A. Booth Referee #2) Section 2.1 (and throughout): You variously refer to your seismic lines by number, or by the source acquisition method. I found this very confusing, trying to remember what method was used on what line, and would prefer that you stick to the numerical reference throughout. The table usefully informs what source was used anyway.

(Authors) We checked the text and kept the numerical reference when mentioning the seismic lines.

(A. Booth Referee #2) P5L25: No need to say "reflected sound waves": if they are transmitted as sound waves, they'll come back as sound waves!

(Authors) The sentence has been modified.

(A. Booth Referee #2) P6L16: What velocity from the 800-4000 m/s range did you settle on? It doesn't seem to be listed anywhere.

(Authors) For lines 2–2b, 6 and 7 we used 1400 m/s in the first 150 m depth and 1800 m/s from 150 m depth to the end of the section. So the velocity changed gradually within this range. This information has been added to the text for these lines, and also for the wireless part in line 5–5b (1300 m/s in the first 75 m depth and 3100 m/s from 75 m depth to the end of the profile).

(A. Booth Referee #2) P7L3: Why the different mute definition for the wireless data?

(Authors) We used a different mute function for the wireless data than for the cabled data because the first option (top mute filter based on the picked first breaks) did not work so well as for the other lines and the cabled part of line 5–5b. Then we tested the surgical mute which worked better for the wireless data.

(A. Booth Referee #2) P7L23: What is this absolute value of error with respect to? Give it as a fraction of the typical target depth?

(Authors) We modified the sentence and now it is: 'This value was justified based on the available borehole data for depth calibration. An error on the order of 1-3 m depth can still be expected, which corresponds to e.g. a 1.3-4% error for a target depth of 75 m'.

(A. Booth Referee #2) P7L25: To help with the interpretation, it might be worth tabulating the expected response of the different geologies you interpret in each geophysical

dataset. Even just listing the range of seismic velocities and resistivities you might expect would help your data description.

(Authors) We thank the reviewer for this interesting suggestion. A table listing seismic velocities and resistivities for every material has been added to the manuscript.

(A. Booth Referee #2) P8L5: Are you implying that the borehole is 0.02 m, or 0.02 *km* away from the seismic line? If it really is 0.02 m, then it hardly seems worth reporting this, and you could just say that the borehole lies on the seismic line.

(Authors) We modified the text and now we include the sentence 'the borehole lies on the seismic line'.

(A. Booth Referee #2) P8L7: The interpreted faults are not really clear, and it seems an over-interpretation particularly since refraction static corrections were not applied. Could near-surface anomalies be the origin of the discontinuities and misalignments that you claim? In any case are the faults and damage zones critical to your model? It seems to me that you could be much more tentative in interpreting them, without damaging any parameter in your model.

(Authors) We agree with the reviewer; the discontinuous reflectivity might be related to near-surface anomalies. For avoiding over-interpretation of the seismic data, we re-evaluated all the sections and removed F features where no geological evidence or other evidence was clear.

(A. Booth Referee #2) P8L8: You don't get a lot of reliable ray coverage in the refraction tomography to really talk about the velocities below reflection B1. I agree that your velocities above this horizon are likely reliable, and you do point out that they have velocities consistent with coarse-grained, saturated sediment. However, in general, I find the resistivity data (Figure 5d, Figure 6e) to provide the much more compelling evidence of a bedrock underburden.

(Authors) We reworked the text for making clearer the information given by the P-wave

refraction tomography velocities in lines 2–2b and 5–5b.

(A. Booth Referee #2) P9L2: On what grounds do you interpret a kinematic response from the seismic data? You see dipping horizons, but I don't see how you can say that this represents a slip surface.

(Authors) We interpreted that the sediments seem disturbed below the landslide scar, apparently moving towards the river (we have modified the sentence). It is only an interpretation, we do not specify that there is a slip surface.

(A. Booth Referee #2) P9L6: I would suggest that it is beyond the capability of travel-time inversion to resolve boulders, as you claim here. I might expect that they could appear as diffractions in the seismic section, or high-resistivity anomalies, but I don't believe that the tomography would be sensitive to them. Furthermore, this over-interpretation doesn't actually appear to influence any parameterisation of your model, so the paper wouldn't be damaged if you said that your tomography has some unexplained velocity artefacts.

(Authors) We followed the reviewer's advice and modified the text accordingly.

(A. Booth Referee #2) P11L20: You suggest that the seismic data shows a higher-resolution delineation of the bedrock/sediment contact, but you wouldn't be able to make this interpretation if it wasn't for the sum total of your datasets! It therefore seems unnecessary to make this assertion when you draw on inferences from all of your data – it doesn't matter which is best! Indeed, this whole section could be considered for removal as it's not clear to me that you are presenting a different hypothesis to one that has been previously postulated. It will always be the case that the use of multiple data sources leads to an improved interpretation.

(Authors) We have reworked this part of the Discussion and removed several parts as the reviewer suggests.

(A. Booth Referee #2) In the interpretation of Figure 9, you correctly point out in the

main text that you are prone to mistaking multiples for genuine reflections. You appear to avoid multiples, except (potentially) for the interpretation between âĹij800-2500 m in Figure 9b. Can you be sure that this hasn't been misinterpreted? Also, the inset figures here add very little: the data look very fuzzy, so much so that the logs don't appear to correlate with anything at all.

(Authors) After careful consideration we removed this channel from the interpretation. We removed the inset figures too.

(A. Booth Referee #2) There are potentially too many figures in the paper, and 12 and 13 could be ear- marked for removal as they're not very clear partly because of the limited quality of the seismic data. Could they be moved into supplementary material instead? Equally, once the interpretation is streamlined, I don't think that all the seismic lines need to be included.

(Authors) We followed this advice and now figures 12 and 13 are included the supplementary materials. We have also moved the figures corresponding to lines 6 and 7, and the single-channel river seismic data to the supplementary material.

(A. Booth Referee #2) Some figure captions need to be reduced in length, typically those relating to the seismic lines (Figures 5,6,8).

(Authors) The length of the captions has been reduced.

---

## Author Response (AR2)

**Responses to the reviewer' comments:**
**Subsurface characterization of a quick-clay vulnerable area using near-surface geophysics and hydrological modelling**

Silvia Salas-Romero[1*], Alireza Malehmir[1], Ian Snowball[1] and Benoît Dessirier[1,2]

[1]Department of Earth Sciences, Uppsala University, Uppsala, 75236, Sweden
[2]Department of Physical Geography, Stockholm University, Stockholm, 10691, Sweden

*Correspondence to:* Silvia Salas-Romero (silvia.salas_romero@geo.uu.se)

We thank A. Booth for reviewing once more our manuscript and for the final comments. We have addressed all the specific comments in our revised manuscript, as detailed below.

**(A. Booth Referee)**
a.d.booth@leeds.ac.uk

P2L22: Start a new paragraph here.

(Authors) We followed the reviewer advice.

**(A. Booth Referee)**
P3L1: "for testing whether an", rather than "for testing that an".

(Authors) The text has been modified.

**(A. Booth Referee)**
P3L17: "surface at a the landslide scar" ... 'the', or 'a' – choose one!

(Authors) The text has been modified.

**(A. Booth Referee)**
P5L25: "provide" rather than "allow to produce".

(Authors) The text has been modified.

**(A. Booth Referee)**
P7L3: confusing use of "both". Do you mean for each line part?

(Authors) Yes, we meant for the cabled geophone and the wireless parts. We modified the text for clarifying.

**(A. Booth Referee)**
Title of Section 5.1: While these are interpretation of data along your "Land Seismic Lines", the interpretation isn't exclusively of land seismic. I suggest you term this "Onshore datasets", and then Section 5.2 "Offshore datasets"?

(Authors) We followed this advice and modified the text accordingly.

**(A. Booth Referee)**
P8L8: I would start this paragraph with the reference to Table 3, which currently appears at the end of it.

(Authors) We followed this advice and now Table 3 is mentioned at the beginning of the paragraph.

**(A. Booth Referee)**
P8L14: Add "values" after "high".

(Authors) The text has been modified.

**(A. Booth Referee)**
P8L17: do you need spaces either side of your hyphen? Such that the text reads "... leached clay deposits - potential quick clays"?

(Authors) As we are not sure what is the correct use of this hyphen in the journal rules, we decided to not let any space on either side of the hyphen and be consistent throughout the text.

**(A. Booth Referee)**
P9L1: I think "and" should be "an"?

(Authors) We meant that below the landslide scar, there are displacement and oblique translation of some reflections. We added a comma for clarity.

**(A. Booth Referee)**
P9L12: "do not delineate" ... not sure exactly what you mean here.

(Authors) We modified the text for clarifying the meaning of this sentence.

**(A. Booth Referee)**
P10L2: "show", not "shows".

(Authors) We checked this in the text and the verb is correctly written. If the reviewer was meaning for the verb in P10L1, we modified the text accordingly.

**(A. Booth Referee)**
P5L27: Link the colour in the greyscale to the anticipated contrast, e.g., do you expect the coarse materials or the fine materials to have the higher density?

(Authors) We modified the text in order to accommodate this comment.

**(A. Booth Referee)**
P9L13-17: In the figure, your white dashed line shows the depth at which the RMT data are unreliable. Unreliable in what sense? Because you suggest that they better-delineate the position of B1 compared to the ATEM data but, if the figure is to be believed, then the ATEM data should be more reliable.

(Authors) We modified, as mentioned before, the sentence related to the delineation of B1. We modified the text in the captions of Figs. 5 and 6. "Reliable" was not a good choice for describing the model; above the thin white dashed line the model is of higher confidence but that does not mean that below the white line the model cannot be trusted, especially since the resistivity structures coincide well with the other geophysical data.
We mentioned the meaning of the white dashed line in the RMT results in the paragraph about Figure 5 (P8L12-15).

**(A. Booth Referee)**
You might also wish to consider explicitly mentioning the RMT/ATEM data as part of the background review. You do mention resistivity data in P3L10, but I still struggled a bit to find where they came from when it came to Figures 5 and 6.

(Authors) The text has been modified for including the RMT/ATEM mention in P3L10.

[revised manuscript text omitted]